# A cell-and-plasma numerical model reveals hemodynamic stress and flow adaptation in zebrafish microvessels after morphological alteration

**Swe Soe Maung Ye** *, **Li-Kun Phng** *

Laboratory for Vascular Morphogenesis, RIKEN Center for Biosystems Dynamics Research (BDR), Kobe, Japan

* swesoe.maungye@riken.jp, swesoe@gmail.com (SSMY); likun.phng@riken.jp (LKP)

## Abstract

The development of a functional cardiovascular system ensures a sustainable oxygen, nutrient and hormone delivery system for successful embryonic development and homeostasis in adulthood. While early vessels are formed by biochemical signaling and genetic programming, the onset of blood flow provides mechanical cues that participate in vascular remodeling of the embryonic vascular system. The zebrafish is a prolific animal model for studying the quantitative relationship between blood flow and vascular morphogenesis due to a combination of favorable factors including blood flow visualization in optically transparent larvae. In this study, we have developed a cell-and-plasma blood transport model using computational fluid dynamics (CFD) to understand how red blood cell (RBC) partitioning affect lumen wall shear stress (WSS) and blood pressure in zebrafish trunk blood vascular networks with altered rheology and morphology. By performing live imaging of embryos with reduced hematocrit, we discovered that cardiac output and caudal artery flow rates were maintained. These adaptation trends were recapitulated in our CFD models, which showed reduction in network WSS via viscosity reduction in the caudal artery/vein and via pressure gradient weakening in the intersegmental vessels (ISVs). Embryos with experimentally reduced lumen diameter showed reduced cardiac output and caudal artery flow rate. Factoring in this trend into our CFD models, simulations highlighted that lumen diameter reduction increased vessel WSS but this increase was mitigated by flow reduction due to the adaptive network pressure gradient weakening. Additionally, hypothetical network CFD models with different vessel lumen diameter distribution characteristics indicated the significance of axial variation in lumen diameter and cross-sectional shape for establishing physiological WSS gradients along ISVs. In summary, our work demonstrates how both experiment-driven and hypothetical CFD modeling can be employed for the study of blood flow physiology during vascular remodeling.

**Data Availability Statement:** All relevant data are within the manuscript and its Supporting information files. The Lattice-Boltzmann and discrete Erythrocyte deformation transport model

used to analyze blood flow and stress in microvascular networks is available in Zenodo (https://doi.org/10.5281/zenodo.7960906).

**Funding:** This work was funded by Japan Society for the Promotion of Science to LKP (22H02624 and 22H05168) and SSMY (JP20K20190). LKP received salary from RIKEN Center for Biosystems Dynamics Research. SSMY was supported by a fellowship from the RIKEN Special Postdoctoral Researcher Program. The funders had no role in study design, data collection and analysis, decision to publish, or preparation of the manuscript.

**Competing interests:** The authors have declared that no competing interests exist.

## Author summary

During angiogenesis, new blood vessels grow into larger networks to expand the cardiovascular system that delivers nutrients to tissue and organs vital for the organism's survival. How the forces of blood flow guide the initial stages of network expansion can be studied with zebrafish embryos due to their optical transparency. However, a key challenge is quantifying the forces of blood flow, which cannot be easily measured from experiments. For this, we have employed computational fluid dynamics (CFD) modeling to predict blood flow forces based on vessel network geometries and blood flow parameters imaged from zebrafish experiments. By combining CFD with experiments on normal zebrafish and zebrafish with genetically transformed blood vessels, we studied how blood flow and its associated forces adapt to such changes. Our analysis highlights that force gradients and distributions in a network follow closely the flow distribution patterns that are in turn adapting to the network geometry and patterning. Through this work, we have demonstrated CFD as a useful quantitative tool for vascular biologists who may employ CFD to complement existing experiment techniques in order to construct or support mechanistic theories of blood flow regulation and vessel remodeling during angiogenesis.

## Introduction

The cardiovascular system is essential for embryonic development, tissue growth and homeostasis in adults by providing sustainable oxygen, nutrients and hormones to tissues. It also aids in the removal of metabolic waste. Angiogenesis, which describes the process of blood vessel network expansion from preexisting vessels, is a topic of active research due to the physiological importance of early stage cardiovascular system development in the survival of a growing organism. While many animal models have been employed for the purposes of such studies [1,2], the zebrafish embryo has been one of the most prolific models for angiogenesis research. This can be attributed to the zebrafish's rapid maturation time, low cost and relative ease of animal husbandry, extensive taxonomy of genetic variants for a multitude of vascular disease representation [3] and most importantly, the relative ease of blood flow and vessel morphology quantification through non-invasive microscopic imaging.

The role of blood flow in vascular embryo development is a topic of growing interest. While a functional cardiovascular system for directional convection of blood is necessary in an adult zebrafish, dysfunction of this system is not fatal for embryos in their first week of existence [4]. Indeed, in the early stages of organ and tissue development, cutaneous delivery of oxygen and nutrients from environmental sources puts less significance on the oxygen delivery function of a vascular network [5]. Despite this, the onset of blood flow at early embryonic stages suggests that, in addition to a nutrient delivery role, blood flow provides instructive cues in the patterning of vascular networks. For example, Notch activation by differing blood flow levels amongst inter-connected vessels regulates endothelial cell (EC) migration involved in determining a vessel's arterial or venous fate in a network [6,7]. In vascular remodeling of plexus connections, vessel pruning occurs primarily in vessels that have lower blood perfusion in comparison to adjacent vessel connections [8,9]. In contrast to pruning, vessels with low perfusion and wall shear stress (WSS) are targeted by intussusception, which expands network connections in the caudal vein plexus by splitting these vessels via trans-capillary tissue pillar formation [10]. During transcellular lumen formation of angiogenic vessels, apical membranes of ECs are exposed to the expansionary forces of blood pressure which drive the lumen invagination process via a tightly regulated process of

tension-induced meta-instability in the apical membrane [11]. Blood flow has also been found to contribute to dorsal longitudinal anastomotic vessel (DLAV) morphogenesis, where blood flow abrogation has been demonstrated to inhibit anastomosis and formation of contralateral plexus connections [12].

These recent studies contribute to the growing narrative that blood flow directs a multitude of cellular events that contribute to angiogenesis and vascular remodeling through nuanced and separate mechanisms. As such, the measurement of hemodynamics in zebrafish is a subdiscipline of growing interest and application within blood vascular development. With confocal fluorescence microangiography and ultramicroscopy, high resolution 3-D geometries of the blood lumen network can be obtained [13–15]. Likewise, cardiovascular performance parameters such as blood flow velocity and flow pulsatility can be measured by imaging techniques such as particle tracking velocimetry [16,17], laser-scanning velocimetry [18] and optical tweezers [19]. However, high precision maps of the lumen WSS and distribution of the blood pressure within the network are parameters that cannot be directly measured. These hemodynamic forces that are essential to understanding EC mechanotransduction and vascular morphogenesis require mathematical models for their calculation from the experimentally obtained flow and geometry data. While there are coarse-grained techniques to assess hemodynamic forces [17], computational fluid dynamics (CFD) models subscribing to Navier Stokes prescription of flow physics has been regarded as the gold standard for mechanistic accuracy and precision. Outside of zebrafish hemodynamics, CFD models employing red blood cell (RBC) and plasma representation of blood transport dynamics have become the expected standard for model prescription in microhemodynamics [20–24]. Conversely, many CFD studies on zebrafish hemodynamics have typically avoided multiphase representation of blood microrheology due to their focus on larger vessels and fluid chambers such as the heart [25–27]. Others have applied a mixed multi-scale scheme of continuum blood prescription at network level to feed boundary flow conditions into the single vessel scale model that prescribe a two-phase RBC and plasma representation of blood [28]. Some CFD studies have omitted RBCs from their blood flow model entirely [29] or limited the number of flowing RBCs to represent only extremely low hematocrit conditions [30]. These coarse-grained modeling approaches are undertaken possibly due to the complexities of the highly frequent inter-RBC collisions and high computational costs associated with deformable RBC simulations. However, the lack of an explicit consideration of RBCs poorly characterizes blood flow in microvascular networks since RBC dynamics directly modulate blood viscosity at micron length scales [31]. The influence of RBCs on blood rheology is particularly important at flow-partitioning bifurcations where plasma and RBCs do not follow symmetric behavior in their flow division [32,33]. Furthermore, the positional asymmetry of the core RBC flow trajectory after bifurcations can heighten the lumen WSS downstream [34,35].

Given the importance of RBC phase representation in blood microrheology, we developed an RBC and plasma CFD model for the study of hemodynamic stress distribution in developing zebrafish. In this study, we aim to demonstrate the usefulness of the CFD approach by analyzing flow and stress alteration patterns in the zebrafish trunk vascular network in response to alterations in hemorheology and network morphology. Additionally, we discuss the process of integrating flow data from experiments into the boundary conditions employed in the CFD model. Essentially, we demonstrate how careful examination and validation of CFD boundary conditions are necessary to properly understand the intimate relationship between flow physiology and compensatory responses to hemorheological or morphological alterations.

## Results

Using microangiography and fluorescence confocal microscopy, we reconstructed a high resolution 3-D *in silico* model of the blood vessel lumen network geometry at the anterior region of the caudal plexus in the zebrafish trunk at 2 days post fertilization (dpf). This plexus comprises 5 intersegmental vessel (ISV) contralateral pairing units, the caudal artery (CA) and the caudal vein (CV) (Fig 1A). The geometry model preserved topological features of the physiological network such as variations in diameters along the vessel axes and irregularity of the lumen cross-sectional profile. To maintain the hematocrit level, we recycled RBCs within periodic boundary faces (two ends of the domains) of the reservoir domains at the anterior of the CA and posterior ends of the caudal vein CV (dashed box domains with cell indices < 344 in Fig 1B). These 3 reservoir domains feed RBCs into the simulation domain by making copies of the recycled cells whenever they exited the ends of the reservoir (see also S1 Movie). Flow at the two periodic faces of each reservoir domain were set by copying the velocity of the adjacent boundary faces in trunk network domain (Fig 1B: $v_{R1} = v_{CA1}$, $v_{R2} = v_{CV1}$, $v_{R3} = v_{CV2}$) where pulsatile pressure boundary conditions (BCs) were input (Fig 1A). Details of hematocrit recycling approach can be found in our earlier work [34]. In order to analyze only developed RBC flows in the network, we initialized RBC and plasma flow from an empty network to eventually fill up to 14% hematocrit (Ht, ratio of total RBC volume to lumen volume) in the CA and 12% Ht in the CV. This was done by periodically repeating RBCs in the 3 reservoir domains to reach the targeted hematocrits (54 RBCs recycling in the CA reservoir, 71 and 14 RBCs in the two respective CV reservoirs). Flow was allowed to develop for 5s before a further 10 cardiac cycles of flow was simulated for the network flow and stress analysis. A key inclusion in our CFD model was the dynamics of the RBC phase. By employing a short-range repulsion force scheme, we could perform RBC-to-RBC contact and collisions with the vessel lumen wall with high regularity and in a robust manner that ensured RBC mesh did not collapse or inter-penetrate due to excessive deformation (Fig 1C, S2 Movie)–for a detailed explanation on this feature, please refer to Models and Methods. Material properties of the RBC and contact dynamics parameters used in the simulations are summarized in Table A of S1 Text.

To represent the pulsatile flow in the trunk network, we employed pulsatile pressure BCs at the anterior and posterior faces of the CA, CV and dorsal longitudinal vessel (DLAV) for a total of 7 pressure (*P*) input BCs (Fig 1A). Notably, we could not apply flow inlet BCs based on the experimental measurements. This was because the full 2-D distributions of the cross-sectional velocity required for flow-input BCs at the anterior and posterior grid points in the LBM model is not measurable by RBC-tracking in our experiments which only provided the lumen centerline velocities. Critically, the relatively large size of RBCs compared to the lumen diameters prevented particle tracking velocimetry from building radial distributions of flow velocities in a lumen. Furthermore, while the cross-sectional velocity distribution is known to adopt a blunt (plug-like) flow profile around the lumen centerline velocity, non-circular lumen cross-sections in the network geometry make it difficult to define analytical velocity distributions as functions of the lumen radius. We also cannot determine the degree of bluntness *a priori* as this itself is the subject of evaluation by CFD which assess the rheological dynamics of cellular interactions in the blood mixture (discussed in the next subsection). The input pressure BCs used in our CFD model are given by the following formulation:

$$
P = \begin{cases} P_{diastole} + P_{range}\sin^2\left(\dfrac{\pi\omega}{0.46}\right) & if\,\omega \leq 0.23 \\ P_{diastole} + 0.5P_{range}\left[1 + \sin\left(\dfrac{2\pi(\omega + 0.54)}{1.54} - \pi/2\right)\right] & otherwise \end{cases}
\tag{1a}
$$

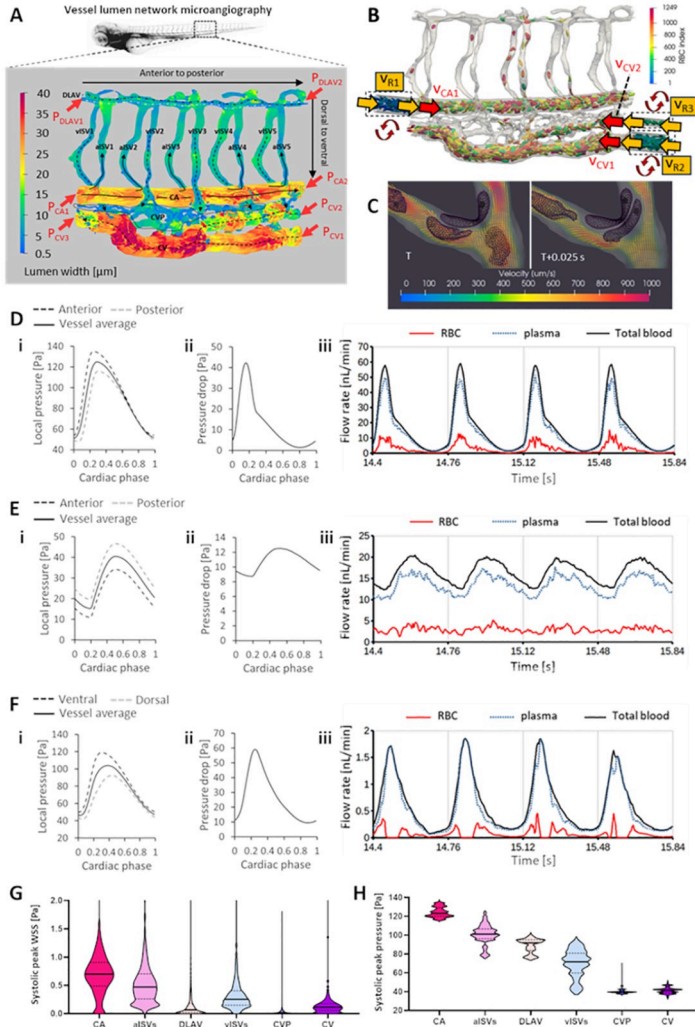

**Fig 1. Development of the cell-and-plasma 3-D computational fluid dynamics (CFD) model. (A)** Morphology of zebrafish trunk network in the caudal vein plexus region obtained using microangiography and confocal microscopy. Shown are the different vessel types of varying diameter (CA: caudal artery, CV: caudal vein, aISV and vISV: arterial and venous intersegmental vessels, DLAV: dorsal longitudinal anastomotic vessel, CVP: caudal vein plexus). Red arrows indicate the 7 locations where pulsatile pressure input boundary conditions are specified. **(B)** Red blood cell (RBC) hematocrit in the simulation domain controlled and maintained by recycling cells in the 3 dashed-box domains (RBC indices < 344), see S1 Movie. Yellow arrows indicate reservoir velocity periodic boundary inputs copied from the adjacent region (red arrow) in the trunk network. **(C)** Collision of RBCs with vessel walls and with one another as part of the explicit consideration of blood dynamics in the CFD model (S2 Movie). **(D and E)** Pulsatile pressure boundary conditions defined at the anterior and posterior ends of the CA (Di) and CV (Ei) and the resulting pulsatile pressure drop across the CA (Dii) and CV (Eii) that drives the pulsatile flow of RBC and plasma blood phases in the CA (Diii) and CV (Eiii). **(F)** Pulsatile pressures arising at the dorsal and ventral ends of aISV3 (i) due to oscillating pressure inputs at anterior and posterior ends of the CA and CV, and the resulting pulsatile pressure drop across aISV3 (ii) that drives the pulsatile flow of RBC and plasma blood phases in aISV3 (iii). **(G and H)** Hierarchical stratification of WSS (G) and blood pressure (H) levels in the network predicted by CFD; see S1 Movie for spatial maps of oscillating WSS and pressure.

$$P = \begin{cases} P_{diastole} + P_{range}\sin\left(\dfrac{\pi\omega}{0.6}\right) & if\,\omega \leq 0.3 \\[2em] P_{diastole} + 0.5P_{range}\left[1 + \sin\left(\dfrac{2\pi(\omega + 0.4)}{1.4} - \pi/2\right)\right] & otherwise \end{cases} \tag{1b}$$

$$where\,\omega = \frac{\left(t - \omega_{lag}T\right)mod(T)}{T} \tag{1c}$$

In Eq (1a) and (1b), $P_{diastole}$ is the diastolic minimum pressure, $P_{range}$ is the width of the pressure variation from diastole to systolic peak and $\omega$ is the cycle phase in cardiac oscillation given by Eq (1c). Following this formulation, we set the systolic (rising $P$) phase to be from $\omega = 0$ to $\omega = 0.23$ for the 2 CA and 2 DLAV input BCs (Eq (1a)); the 3 input BCs for the CV have lower pulsatility and longer systolic phases from $\omega = 0$ to $\omega = 0.3$ (Eq (1b)). Phase lag of the $P$ oscillation in a particular pressure input location is given with respect to the phase at the anterior end of the CA, and is applied in Eq (1c) with the lag coefficient ($\omega_{lag}$). $T$ is the period of one cardiac oscillation. All the oscillating $P$ input BC parameters employed in the wild-type (WT) and other network phenotypes models in this paper are summarized in Table B of S1 Text.

Consequently, by employing pulsatile pressure boundaries at the anterior and posterior ends of the CA, CV and DLAV, RBC and plasma phases developed the local flow profiles in accordance with the time-dependent pressure distribution around the network (Fig 1D to 1F). Larger pressure oscillations and pressure drops in the CA (Fig 1Di and 1Dii) compared to CV (Fig 1Ei and 1Eii) produced more pulsatile and higher systolic peak flow-rates in the CA (Fi. 1Diii) than in the CV (Fig 1Eiii). However, the time-averaged flow rate for RBC (2.86, 2.94 nL/min); and plasma (13.3, 13.3 nL/min) phases in both CA and CV were similar, indicating mass conservation in the CFD models. Oscillating pressures established along the dorsal and ventral ends in the intersegmental vessels (ISVs) that connect CA to DLAV (shown for aISV3 in Fig 1Fi) and DLAV to CV determined the oscillating pressure drop (Fig 1Fii) that drives fluctuating RBC and plasma flow rates in the ISVs (Fig 1Fiii).

We ensured that the CFD flow results were a reasonable match to our experimental reference [17] in terms of blood flow peak centerline velocities in the lumen center and the time-averaged discharge hematocrit (Hd, ratio of average RBC flow against total blood flow) in the various vessel types (Table C of S1 Text). With regards to the method of obtaining the pressure settings reported in Table B of S1 Text, the primary guiding principle was first to ensure a mass conservation in the network in that the time-averaged flow rates for both plasma and RBC phases matched between the CA and CV. Next, we incrementally shifted pressure ranges over the initial 5s of flow development to match CFD-predicted peak velocities to peak velocities reported in our experimental reference [17]. We systematically reduced or increased the anterior-posterior trunk pressure inputs to minimize the peak velocity discrepancy between CFD results and our experimental reference to levels below 25% (Table C of S1 Text). For example, if the CFD velocities in the CA were 30% higher than the experiment levels, the anterior to posterior pressure drop in the updated pressure input for the two ends of the CA would be reduced by 30%. Further modification to the pressure inputs involve modulating the pulsatility range by modifying $P_{diastole}$ and $P_{range}$ while maintaining the time-averaged $P$ in Eq (1). Additionally, we compared the regional blood flow velocities in the CA, CV and ISVs to levels observed within our flow-imaging experiments conducted for the network phenotypes in this study (Tables D–F of S1 Text). Pressure settings for the phenotype networks were based on the

experiment trends observed (discussed in Results and Discussion sections). All vessel flow and lumen diameter analysis performed for imaging experiments were taken from the pooled-averages of vessels in the anterior caudal trunk region (just behind the yolk extension; see Figs A–E of S2 Text) for each respective experiment group. Summarizing the CFD output for the WT model, we obtained spatiotemporal maps of wall shear stress (WSS) and lumen blood pressure (see S1 Movie). From these maps we verified the establishment of WSS hierarchy among vessel types, from high to low (Fig 1G): CA, aISVs, venous intersegmental vessels (vISVs), CV, DLAV and the caudal vein plexus capillaries (CVP). As blood flows from high to low pressure regions, the hierarchy of blood pressure levels follows this arterial to venous flow direction. The blood pressure hierarchy given from highest to lowest was CA, aISVs, DLAV, vISVs, CVP and CV (Fig 1H).

## Hematocrit modulates blood pressure in a microvascular network

In microvessels, RBC deformation and interactions with vessel walls can directly modulate blood viscosity. We set up a CFD model of a straight vessel segment (30 μm in diameter and 150 μm in length) where we varied RBC concentration (Fig 2A, S3 Movie) and RBC-to-RBC interactions (Fig 2B, S4 Movie for 10% Ht and S5 Movie for 20% Ht) to study their resulting effects on blood viscosity. Increasing the RBC concentration from 10% hematocrit (Ht) to 20% Ht resulted in a decrease in blood flow velocity (Fig 2Ai and 2Aii) and a 30% increase in effective viscosity of blood (Fig 2C). When RBCs were staged in the model to undergo human-physiological levels of aggregation at 1 μJ/m2 [36] attraction strength, we observed clumping of RBCs into rouleaux clusters in the center of the lumen. Correspondingly, a blunting of the cross-sectional flow velocity profile was observed (Fig 2D). The rouleaux formation lowered effective blood viscosity in the vessel by 3% and 11% in the 10% and 20% Ht scenarios, respectively (Fig 2C). Essentially, in this test scenario the pressure drop across the vessel segment has been maintained at 9.6Pa while blood viscosity was modulated, as similarly performed by Pries and colleagues with glass microcapillary flow experiments under constant pressure drop [37].

While RBC aggregation is not a reported phenomenon in zebrafish, hematocrit level variation is typical of embryos at 2 dpf [17]. Furthermore, hematocrit manipulation is a common practice among researchers where *gata1* knockdown zebrafish models with diminished RBC count [4] are used to study the effects of blood viscosity reduction on angiogenesis and vessel remodeling [38]. It is widely reported that *gata1* knockdown leads to a systemic decrease in blood viscosity and thereby reduced WSS in blood vessels [39]. While this mechanism for WSS reduction applies to blood vessels with high RBC perfusion such as the heart [39] and the caudal vein plexus (CVP) [28], vessels like some ISVs with already low RBC perfusion are not expected to have significant lowering of blood viscosity from a systemic hematocrit reduction. Instead, a redistribution of pressure in the entire network is expected and how this determines the adapted WSS levels in such ISVs is not clear.

To address the uncertainty over network pressure gradients and their adaptive response to hematocrit reduction, we performed experiments where 7 *gata1* morpholino injected and 8 control morpholino injected embryos were imaged between 53–57 hpf (Fig A of S2 Text and Table D of S1 Text). We obtained a continuum range of hematocrit and studied their correlative effects on flow and vessel morphology (Fig 3A to 3F). Here, the relative CA hematocrit is the RBC-count divided by average lumen cross-sectional area (for each embryo) and normalized against the maximum level seen in control morpholino injected embryos (Control MO#3 in Fig Aviii of S2 Text). Regression *t*-tests indicated that both aISVs and vISVs had reduced vessel diameters in response to the hematocrit reduction (Fig 3A and 3B). The CA and CV on the other hand maintained similar vessel diameters (Fig 3C and 3D). Despite the wide

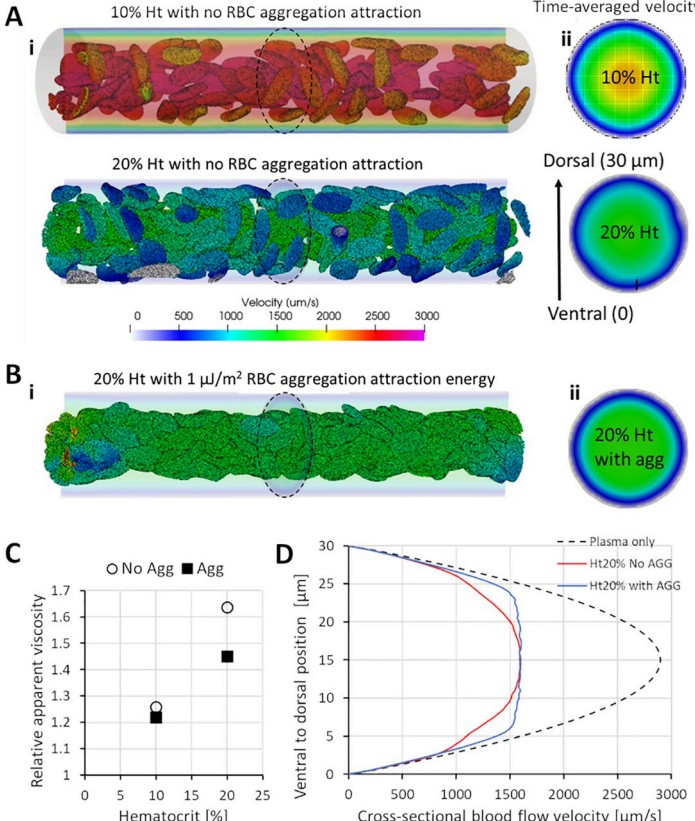

**Fig 2. Effects of RBC hematocrit and aggregation on viscosity and blood flow velocity. (A)** Snapshots of RBC arrangement and velocity driven by steady state 9.6 Pa pressure drop from left to right ends of the vessel segment (30 μm in diameter and 150 μm in length) under 10 and 20% hematocrits (i) and the time-averaged blood velocity plot in the cross-sectional lumen at the middle of the vessel length (ii). **(B)** Snapshots of RBC rouleaux formation and velocity under 20% hematocrits and 1 μJ/m² aggregation levels (i) and the time-averaged blood velocity plot in the cross-sectional lumen at the middle of the vessel length (ii). **(C)** Plot of relative apparent viscosity against hematocrit for blood with and without RBC aggregation. Movies of the four conditions simulated can be viewed in S3–S5 Movies. **(D)** Flow retardation effect of RBCs in blood flow and the flow blunting effects of RBC aggregation on the cross-sectional velocity profile under constant segment pressure drop conditions.

biological variation in heart rate amongst embryos, the level was statistically insensitive to the hematocrit reduction (Fig 3E) across the population of zebrafish examined (n = 15). Importantly, from the RBC velocity and vessel diameter quantification, the estimated flow rate in the CA was found to have been maintained across the wide range of hematocrit reduction (Fig 3F; see Section A of S2 Text for details of hematocrit and flow rate estimation).

Next, we performed CFD simulations following the experiment finding that flow rate was maintained in the CA despite hematocrit variation. 3 models were used for comparison of WSS levels in different vessels with regards to the hematocrit level. WT model employed a physiological RBC hematocrit (14% Ht) in a wildtype lumen network while model NoRBC1 was devoid of RBCs in the same network. Model NoRBC2 was devoid of RBCs in the trunk network where ISV diameters were reduced by 18% in comparison to WT and NoRBC1. The ISV diameter reduction in NoRBC2 follows the regression model fit based on the diameter-hematocrit correlation observed in the experiment (Fig 3A and 3B). Since the blood viscosity was lowered in NoRBC1 and NoRBC2, flow rates in the CA and CV for these two models were maintained at WT levels in the simulations by lowering the CA and CV blood pressures

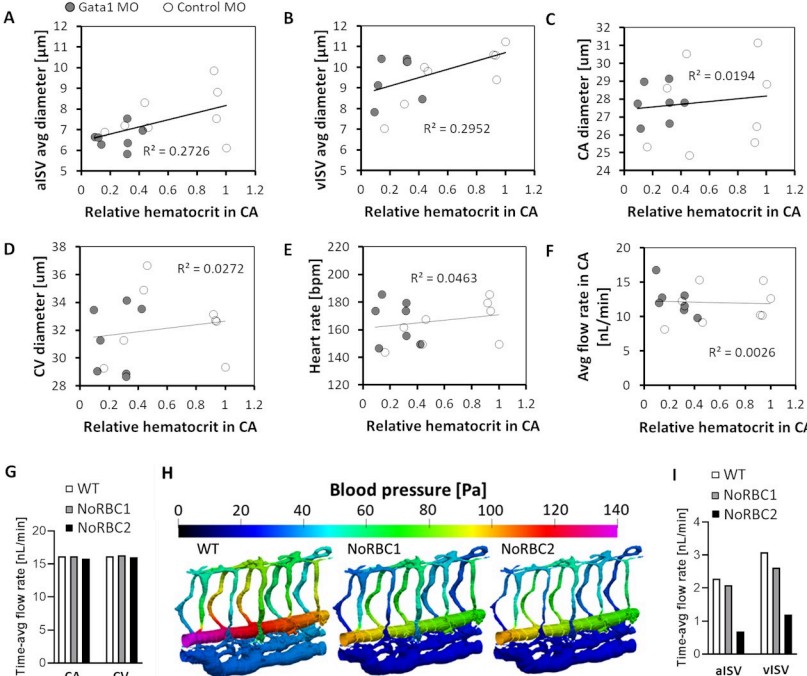

**Fig 3. Systemic alteration in vessel morphology and blood flow in zebrafish with reduced RBC hematocrit. (A to D)** Diameters of aISV: $p = 0.046$ (A), vISV: $p = 0.036$ (B), CA: $p = 0.63$ (C) and CV: $p = 0.56$ (D) as functions of the relative CA hematocrit in *gata1* and control morphants. **(E to F)** Heart rate: $p = 0.44$ (E) and estimated flow rate: $p = 0.86$ (F) as functions of the relative CA hematocrit. Statistical significances of slopes in A to F were determined using regression *t*-test. **(G)** CFD model predictions of blood flow rates in CA (16.15, 16.15, 15.76 nL/min) and CV (16.21, 16.34, 16.06 nL/min) for WT (14% Ht), NoRBC1 (0% Ht) and NoRBC2 (0% Ht and 18% ISV diameter reduction). **(H)** CFD predictions of systolic pressure distribution maps in models WT, NoRBC1 and NoRBC2. **(I)** CFD predicted blood flow rates in the aISVs (-9.1% and -70% for NoRBC1 and NoRBC2, respectively, compared to WT) and vISVs (-15% and -61% for NoRBC1 and NoRBC2, respectively, compared to WT).

employed in WT model (Fig 3H, see also Table B of S1 Text for the pressure inputs). Effectively, this reduced the pressure gradient across ISV networks for NoRBC1 and NoRBC2, thereby resulting in lower blood flow rates in the aISVs and vISVs compared to WT (Fig 3I). The larger reduction in ISV flow rates for NoRBC2 can be attributed to the ISV lumen diameter reduction in addition to the pressure gradient weakening.

In terms of WSS, there was a general lowering of levels across the networks for NoRBC1 and NoRBC2 (Fig 4A). Compared to WT, WSS was reduced by similar levels (26–27%) in the CA and CV for NoRBC1 and NoRBC2 (Fig 4Bi and 4Bii). WSS was about 20% lower in NoRBC1 than WT for both aISVs and vISVs. NoRBC2 saw similar drop in WSS for aISVs and vISVs but by twice the amount as NoRBC1 (Fig 4Biii and 4Biv). While the WSS reduction in the CA and CV can be attributed to the lowered blood viscosity attendant with hematocrit reduction, a different causative principle was responsible for the WSS lowering in ISVs of NoRBC1 and NoRBC2 since the ISV hematocrit reduction from WT levels was marginal (Ht reductions from 2.5% Ht in vISVs and 4.6% Ht in aISVs for WT to 0% Ht in NoRBC1 and NoRBC2). Instead, reduced blood flow rates in the ISVs of these two models drove WSS levels down. Furthermore, applying the ISV diameter reduction adaptation observed in *gata1* morphants in NoRBC2 increased vessel impedance to further reduce blood flow in ISVs compared to NoRBC1, thereby exacerbating the decrease in WSS. Hence, our CFD results suggest that the predicted WSS reduction in ISVs for hematocrit-reduced networks was not due to viscosity

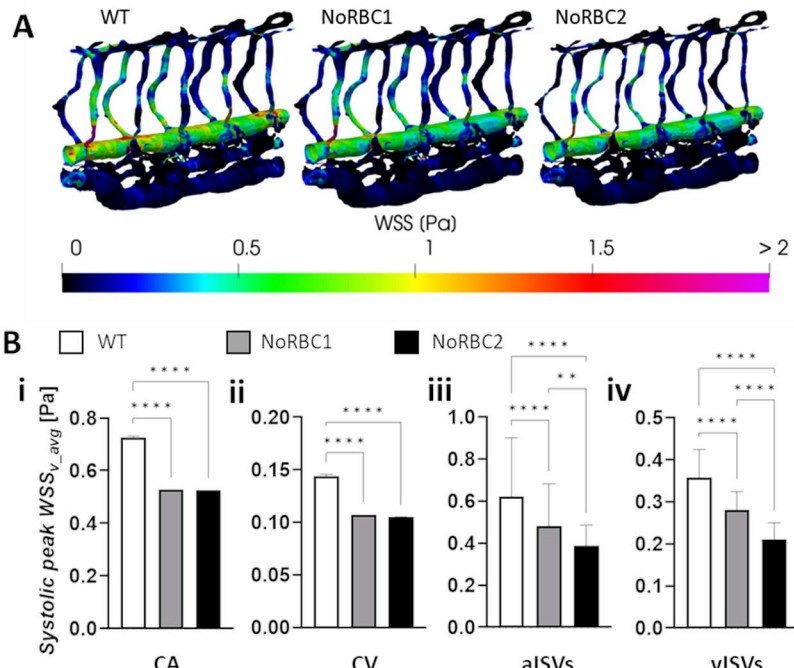

**Fig 4. Decrease in ISV diameter exacerbates the decrease in WSS caused by hematocrit reduction. (A)** Systolic WSS distributions in models WT (14% Ht), NoRBC1 (0% Ht) and NoRBC2 (0% Ht and 18% ISV diameter reduction). **(B)** Predicted systolic WSS levels in the CA (i), CV (ii), aISVs (iii) and vISVs (iv) for the 3 models. Statistical comparisons were performed using one-way ANOVA with Sidak's multiple pair comparisons. **, $p < 0.01$; ****, $p < 0.0001$.

reduction per se but rather due to the compounding effects of the loss in the dorsal-ventral pressure gradient and the apparent adaptation response of ISV lumen narrowing.

## Ventral narrowing of ISV diameter tunes network perfusion and stress gradients

While the application of 3D CFD modeling of hemodynamics in the zebrafish vascular network is not completely novel, a realistic representation of local topologies in the network is a feature often ignored in many CFD models that represent the lumen cross-sectional profile to be smooth and circular [28,29]. Most importantly, the vessel lumen narrows and dilates along the ISV axis at seemingly predestined locations (Fig 1A). Here, we performed parametric modeling of the lumen diameter and lumen cross-sectional shape variation to elucidate how patterns of lumen variation along vessel segments tune network hemodynamics. We generated an idealized set of vessel networks that were of varying degrees of simplifications to the original WT lumen morphology obtained from real geometry (WT). We termed this idealized network set the smooth-geometry model (SGM). In the SGM1 model, dorsal-to-ventral variation in ISV diameters observed in the WT geometry was preserved but skewness in cross-sectional lumen profiles of the WT were simplified to ellipse fits in the SGM1. The SGM2 network likewise preserved the axial variation in diameter but further simplified the cross-sectional profiles into circular-fits, thus losing representation in luminal skewness seen in the WT and SGM1. Finally, the highest degree of geometric simplification of ISVs were performed in the SGM3 where the vessels were sized with constant diameters obtained from the geometric average of segments diameters along the original WT ISVs (Fig 5Ai, 5Aii and 5Aiii).

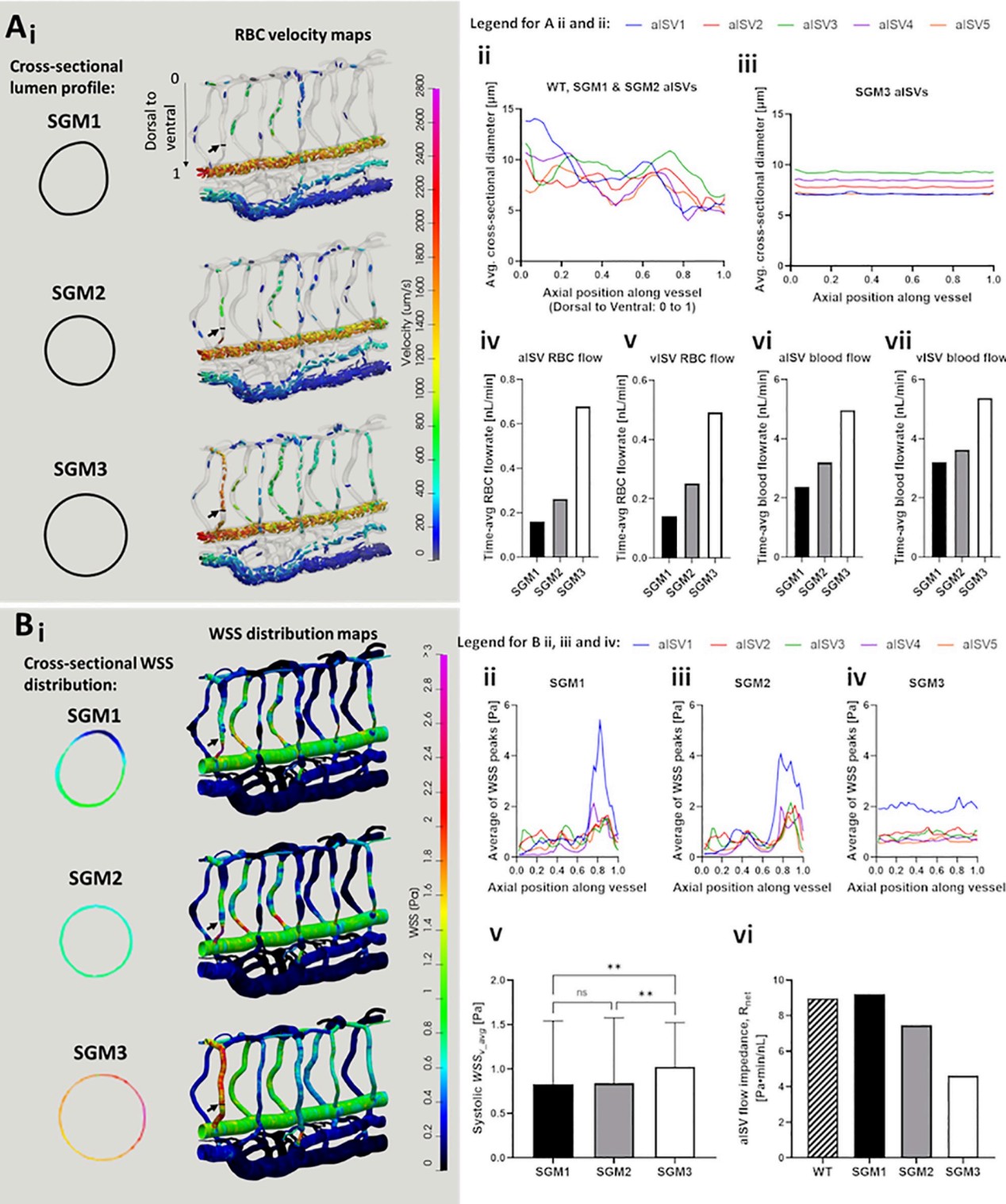

**Fig 5. Simulation predictions of flow and WSS distribution in networks with ISVs of varying diameters and cross-sectional shapes.** (see also S6 Movie). **(A)** RBC partitioning and velocity maps for the 3 SGM networks. Magnified cross-sectional profile of aISV1 lumen at 0.75 dorsal-to-ventral position is indicated for the 3 SGM by the arrows (Ai). The variation in average cross sectional aISV lumen diameter along the dorsal-to-ventral axis observed for WT network that was similarly prescribed in the SGM1 and SGM2 ISVs (Aii). The geometric averaging of lumen diameter applied across ISV axis in the SGM3 network (Aiii). Time-averaged aISV (Aiv) and vISV (Av) RBC flow rates. aISV (Avi) and vISV (Avii) blood flow rates across the 5

aISVs in the SGM models. **(B)** Systolic peak WSS maps for the 3 SGM networks and the magnified cross-sectional WSS distribution in aISV1 lumen at 0.75 dorsal-to-ventral position (Bi). Dorsal-to-ventral plots of systolic WSS in aISVs in the SGM1 (Bii), SGM2 (Biii) and SGM3 (Biv) networks. Comparison of the average systolic WSS (Bv) and the corresponding vessel flow impedance ($R_{net}$) (Bvi) across the 5 aISVs in the 3 SGM networks. Statistical comparisons were performed using one-way ANOVA with Sidak's multiple pair comparisons. **, $p < 0.01$.

We regarded SGM1 as the control model closest to WT, but did not use WT for this comparison exercise. This was because we wanted to set the entire network geometry and the input pressure boundary conditions to be the same between the SGM models except for the modifications made in the ISVs. The most apparent consequence of the progressive geometrical simplification from SGM1 to SGM3 was the increasing RBC perfusion trend in the ISVs (Fig 2Ai and S6 Movie). Firstly, comparing SGM1 and SGM2, the loss of vessel (cross-section) skewness in SGM2 resulted in a moderate increase in the RBC flow rate over SGM1 by 64% for aISVs (Fig 5Aiv) and 77% for vISVs (Fig 5Av) and blood flow rate by 34% for aISVs (Fig 5Avi) and 13% for vISVs (Fig 5Avii). The loss of dorsal-to-ventral lumen size variation in the SGM3 saw 159% (aISV) and 96% (vISV) increases in RBC flow rate and 55% (aISV) and 48% (vISV) increases in blood flow rate over SGM2. Meanwhile, the CA and CV flow were maintained at similar levels between the SGM models, thus indicating that the three models were prescribed with similar flow delivery conditions in the main flow conduits of the network (see Fig F of S2 Text for this data). Hence, it was clear that ventral narrowing had a larger ISV flow reduction effect than caused by lumen cross-sectional irregularity. Furthermore, the reduction disproportionately affected the RBC phase more than it did the plasma.

Owing to the variation in lumen width along the ISVs in SGM1 and SGM2, the distribution of systolic peak WSS were spatially irregular in SGM1 and SGM2. This was in contrast to SGM3 which displayed smooth distributions of almost constant WSS along the entire ISV length (Fig 5Bi). There was a positive dorsal-to-ventral WSS gradient in the aISVs for SGM1 and SGM2 (Fig 5Bii and 5Biii) that was absent in the aISVs of SGM3 (Fig 5Biv). Essentially, aISVs characterized by the simplified constant diameter tubes in SGM3 have very mild shear and pressure gradients (see S6 Movie) along their vessel axis. Such misrepresentation of stress gradients may affect analyses on how flow and stress patterns guide mechanotaxis during endothelial cell migration [40] and polarization [41,42]. Furthermore, the systolic peak WSS levels in the SGM3 model (mean 1.025 Pa) was higher than both SGM2 (mean 0.8384 Pa) and SGM1 (mean 0.8316 Pa) by statistically significant margins (Fig 5Bv). On the other hand, systolic peak WSS level for ISVs was statistically similar between SGM1 and SGM2 despite the increase in RBC flux across the ISV networks in SGM2 over SGM1. Thus, suggesting that vessel skewness might not play a significant role in tuning network WSS. Finally, we calculated the effective flow resistance ($R$) of each aISV in the SGM models and WT by dividing the pressure difference ($P_{drop\_v}$) across the vessel by the flow rate ($Q$) running through the vessel (Eq (2)). The aggregate resistance of aISVs in the modelled network is termed the aISV impedance ($R_{net}$) and is akin to the concept of electrical resistors arranged in parallel (Eq (3)):

$$R = P_{drop\_v}/Q \tag{2}$$

$$R_{net} = \left( \sum \frac{1}{R} \right)^{-1} \tag{3}$$

As shown in Fig 5Bvi, impedance $R_{net}$ for aISVs in the WT (8.98 Pa·min/nL) and SGM1 (9.2 Pa·min/nL) are similar. $R_{net}$ deviates further away from the WT level in the SGM2 (7.47 Pa·min/nL) and SGM3 (4.63 Pa·min/nL).

In summary, we highlight that the dorsal-to-ventral narrowing of aISVs at 2 dpf appears to confer the network with flow and WSS mitigation features primarily by restricting the entry of RBCs into the aISVs. Based on our findings with the SGM simulation sets, capturing the correct stress gradient trends and general WSS levels of the real network geometry requires at least a recapitulation of the diameter variation along vessel axes in the *in silico* network geometry. The recapitulation of cross-sectional skewness and irregularity in the *in silico* lumen model is important for correctly representing vessel flow impedance and RBC flow partitioning in the network.

## Hemodynamic forces in vascular networks of altered lumen size

**ISV lumen size reduction reduces systemic flow.**   After examining the basic features of the WT network geometry, blood rheology and their relationship with WSS and pressure distributions, we applied alterations in network morphology to study the effects of morphological alterations to network flow and stress distribution. We generated *in silico* vascular networks based on vessel morphologies obtained from experimental manipulations. We took the WT as a baseline geometry and modified it in accordance with reported metrics of the lumen geometry observed from confocal images.

The first group of experiment-based networks were the vascular phenotypes obtained from Marcksl1 gain- and loss-of-function experiments. In the case where Marcksl1, an actin bundling protein, is overexpressed specifically in endothelial cells (Marcksl1 OE), the modified network is a lumen bulbous-expansion phenotype [43] compared to WT (Fig 6A vs Fig 6B), where we selected four local regions for lumen volume dilation (dashed yellow circles in Fig 6B): Region 1: vISV2 (+239%), Region 2: aISV3 (+288%), Region 3: DLAV3 (+300%), Region 4: DLAV4 (+272%). In the case of *marcksl1a* and *marcksl1b* double knockout zebrafish (Marcksl1 KO), the modified network is a reduced lumen diameter phenotype [43] (Fig 6C–6E): aISV (-20%), vISV (-23%), DLAV (-21%), CA (-10%), CV (-5%).

We maintained the same input pressures in the Marcksl1 OE model (ML1OE) as the WT model (Fig 6Aii and 6Bii). Since the lumen dilations were localized to a few locations in the ISV network, we did not expect the overall systemic flow to be altered for the Marcksl1 OE network. However, we did not expect the same outcome for the Marcksl1 KO network. Since the lumen reduction was systemically applied across the entire network, the systemic flow in the Marcksl1 KO will likely undergo physiological adaptation. Consequently, we have considered three hemodynamics adaptation scenarios for the Marcksl1 KO network CFD modeling. In Marcksl1 KO model 1 (ML1KO1), the blood pressure was reduced (92 Pa peak systolic blood pressure in ML1KO1 versus 136 Pa in the WT, Fig 6Cii). In the second Marcksl1 KO model (ML1KO2), the blood pressure was maintained to same levels as WT (Fig 6Di and 6Dii). In the third Marcksl1 KO model (ML1KO3), the blood pressure was increased (164 Pa peak systolic blood pressure) to maintain the same CA blood flow rate (16.0 nL/min) as WT (16.2 nL/min) despite the lumen reduction (Fig 6Ei and 6Eii).

We sought to verify which adaptation scenario was most likely by performing experiments where the blood flow in Marcksl1 KO zebrafish (15 embryos) and WT (5 embryos) at 2 dpf (Table E of S1 Text, Figs B–E of S2 Text) were imaged. Through fluorescent labelling of RBCs using *Tg(gata1:dsRed)$^{sd2}$* zebrafish, we measured the RBC flux into the ISV networks. RBC-perfused lumen diameters of the trunk vascular network were obtained by performing maximum intensity time-stack projections of the RBC trajectories in accordance with the procedure described in Ye and colleagues [17]. From these two measurements we could classify the degree of phenotype for Marcksl1 KO zebrafish in terms of RBC perfusion levels $\dot{N}_{RBC,ISV}$ [RBCs/s per ISV] into the ISV network. Representative images of these groups are shown in

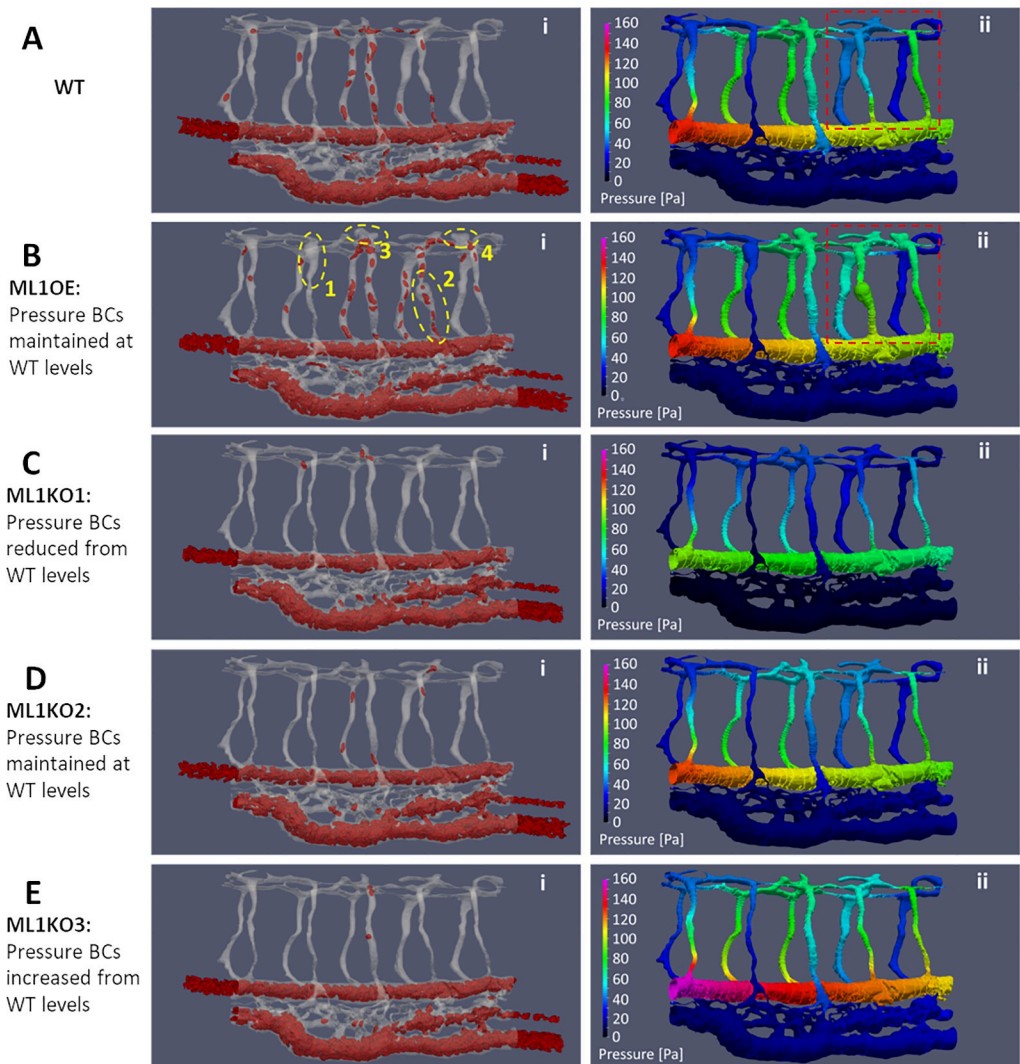

**Fig 6. Hemodynamics adaptation scenarios for Marcksl1 OE and Marcksl1 KO. (A)** WT levels of RBC (i) and systolic blood pressure (ii) distribution, see S1 Movie. **(B)** Marcksl1 OE model (ML1OE) with same blood pressure boundary conditions (BCs) at anterior/posterior input locations as WT. Spatial maps of RBC perfusion (Bi) and systolic blood pressure distribution (Bii) where yellow dashed circles indicate the locally dilated lumen region compared to WT (Bi), see S7 Movie. Dashed red box region in Aii and Bii is the local network region where pressure distribution has been altered between WT and ML1OE. **(C—E)** Marcksl1 KO Model 1 (ML1KO1, C) with reduced blood pressure BCs, Marcksl1 KO Model 2 (ML1KO2, D) with same blood pressure BCs as WT and Marcksl1 KO Model 3 (ML1KO3, E) with increased blood pressure BCs. Spatial maps of RBC perfusion (i) and systolic blood pressure distribution (ii), see S8–S10 Movies.

Fig 7Ai–7Aiii. In the high perfusion group were 4 embryos with $\dot{N}_{RBC,ISV} > 4$. In the moderate (mid) perfusion group were 6 embryos with $4 > \dot{N}_{RBC,ISV} > 1$. In the low perfusion group were 3 embryos with $1 > \dot{N}_{RBC,ISV} > 0$. In the zero perfusion group were 2 embryos with no RBC perfusion into the ISV network.

The experiment results showed that compared to the average WT ISV lumen diameter (D = 5.99 μm), there was a statistically significant reduction in average ISV lumen diameter for the low perfusion group (D = 4.61 μm) and moderate perfusion group (D = 4.91 μm). The average ISV lumen diameter in the high perfusion group (D = 5.80 μm) was statistically similar

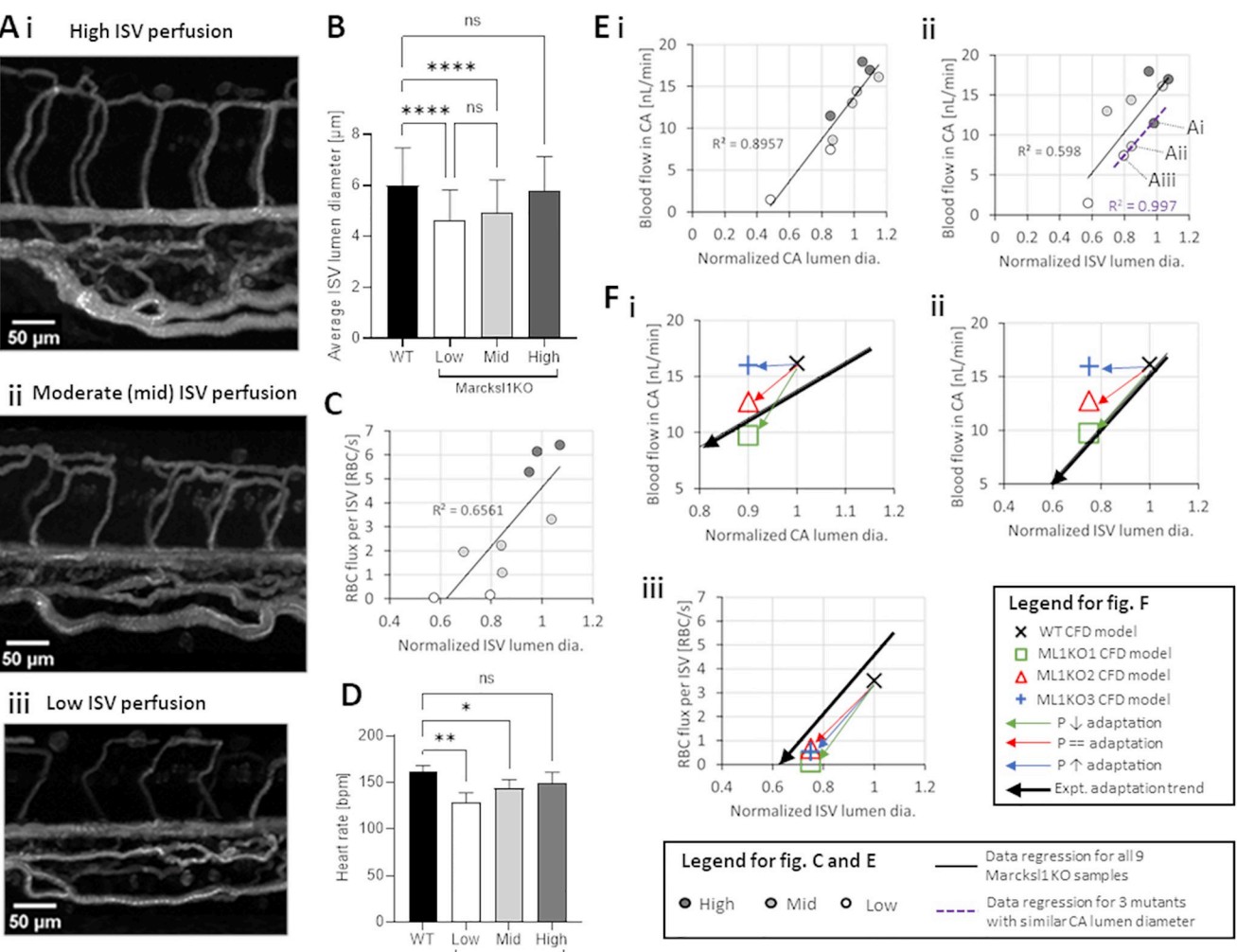

**Fig 7. Characterization of RBC flow in Marcksl1 KO zebrafish with reduced vessel diameters. (A)** Representative maximum intensity time-stack projection images of RBC flow in the ISV networks of 2 dpf Marcksl1 KO zebrafish, from high (i), moderate/mid (ii) to low (iii) perfusion. **(B)** Comparison of average ISV lumen diameters of Marcksl1 KO zebrafish in different perfusion level groups versus WT. **(C)** RBC flux reduction in ISVs in response to ISV lumen diameter reduction (in Marcksl1 KO zebrafish). **(D)** Comparison of heart rate in Marcksl1 KO zebrafish versus WT. Statistical comparisons in B and D were performed using one-way ANOVA with Sidak's multiple pair comparisons. *, $p < 0.05$; **, $p < 0.01$; ****, $p < 0.0001$. **(E)** Experimental observations of flow reductions in Marcksl1 KO mutants. Arterial blood flow reduction in CA in response to reduced CA lumen diameters (i). Arterial blood flow reduction in CA in response to reduced ISV lumen diameters (ii). Purple dashed lines indicate data points from the same embryos in Ai to Aiii, which have similar CA lumen diameter. **(F)** Comparison between the modeled adaptation scenarios (green, red and blue arrows) compared to experimentally observed adaptation trends (black arrow: regression data from E) for CA blood flow against CA diameter (i), CA blood flow against ISV lumen diameter (ii) and RBC flux entering ISVs against ISV lumen diameter (iii).

to WT (Fig 7B). Expectedly, the level of RBC perfusion into ISV networks correlated well with the level of ISV lumen diameter reduction observed across the mutant population (Fig 7C). In terms of heart rate, there was a statistically significant reduction in the average heart rate for the low (129 bpm) and mid (144 bpm) groups compared to WT (162 bpm), but not for the high group (149 bpm) (Fig 7D). As in the case of our earlier observation with *gata1* morphants, we considered the possible link between heart rate and cardiac output and thus estimated the CA flow rate in the mutants. The blood flow rate decreased in accordance with decreases in both CA (Fig 7Ei) and ISV lumen diameter (Fig 7Eii). This indicated that the physiological response to systemic lumen narrowing in mutants was flow reduction in their trunk networks.

We compared these experiment trends to the CFD results in models ML1KO1, ML1KO2, ML1KO3. The adaptive response of flow-rate reduction can be predicted by both ML1KO1 (red arrow path in Fig 7Fi–7Fii) and ML1KO2 (green arrow path in Fig 7Fi–7Fii). An increase in network pressure in order to maintain WT-level CA flow rates as assumed in ML1KO3 is not supported by the experiment. All 3 adaptation models predicted a decrease in RBC flux entering the ISV networks as a result of decreased ISV diameters (Fig 7Fiii). Although the absolute RBC flux levels in ISVs for the simulated WT, ML1KO1, ML1KO2 and ML1KO3 CFD models were lower than their respective experiment averages, the relative reduction from WT levels to Marcksl1 KO levels were similar to the experimental trend (Table F of S1 Text: 80% flux reduction in experiment, 93%, 89% and 91% flux reduction in models ML1KO1, ML1KO2 and ML1KO3, respectively). In summary, from the comparisons of the model trends for CA flow adaptation against the experiment observation, we determined that the flow reduction scenario assumed in models ML1KO1 and ML1KO2 are both likely to happen and the resulting WSS distribution scenario for these two Marcksl1 KO networks were consequently examined.

**ISV lumen size alteration redistributes network pressure and WSS levels.** Before discussing the adaptation scenarios and their impact on WSS distribution in the network, the pressure redistribution effects of lumen size alterations must first be appreciated. In the Marcksl1 OE model (ML1OE), the local dilation of aISV4 (circled region 2 in Fig 6B) correspondingly leads to a weakening of the flow driving pressure difference ($P_{drop\_v}$) in that vessel compared to the WT network (Fig 6Bii vs 6Aii, Fig H of S2 Text). Conversely, a dilation of the DLAV in circled region 4 (Fig 6B) raised the $P_{drop\_v}$ for aISV5 and vISV4 in the ML1OE model. Here the lumen dilations in region 2 and region 4 worked in tandem to lower the local network impedance for aISVs and vISV units 4 and 5 ($R_{net4,5}$ = 47.7 Pa·min/nL for WT and $R_{net4,5}$ = 32 Pa·min/nL for ML1OE; red dashed box in Fig 6Aii and 6Bii). Dilations in circled regions 1 and 3 can be interpreted as inconsequential alterations to the network pressure since they are network regions isolated from network flow. Likewise, the lumen reduction performed in ML1KO1 and ML1KO2 also led to redistribution of network pressures. Like ML1OE, ML1KO2 was staged with the same input pressure boundary conditions (BCs) as the WT. However, only aISV1 and aISV4 maintained $P_{drop\_v}$ similar to WT levels. $P_{drop\_v}$ levels were notably higher than WT levels in aISVs 2, 3 and 5. Hence, it should be noted that due to network alterations between WT and the Marcksl1 OE or KO vasculature, the network pressures have been redistributed and we cannot expect uniformly similar pressure trends between WT and the Marcksl1 OE or KO networks.

While lumen diameter is inversely related to the WSS level, two other hemodynamic parameters positively contribute to WSS in a vessel, 1) the flow-driving pressure difference ($P_{drop\_v}$) and 2) the blood flow rate in the vessel. Both ML1KO1 and ML1KO2 have reduced lumen diameters that promotes WSS increase. Changes to $P_{drop\_v}$ and flow rate either mitigate or further promote WSS increase, and this is where ML1KO1 and ML1KO2 differ. As a result of the reduced pressures in the CA, CV and DLAV, $P_{drop\_v}$ in aISVs for ML1KO1 was statistically similar to the WT level (Fig 8Ai) while $P_{drop\_v}$ in the vISV was lowered (Fig 8Bi: -20%). On the other hand, maintenance of the CA, CV and DLAV pressures at WT levels for ML1KO2 saw $P_{drop\_v}$ levels increase for aISVs (Fig 8Ai: 15%) and maintained at WT levels for vISVs. The aISV and vISV impedances were similar between ML1KO1 and MLKO2, and were raised from WT levels (Fig 8Aii and 8Bii). The increased impedance resulted in blood and RBC flow reductions in both models compared to WT (Fig 8Aiii, 8Aiv, 8Biii and 8Biv). The larger flow reduction in ML1KO1 can be attributed to the lower $P_{drop\_v}$ for ISVs in ML1KO1 compared to ML1KO2. The larger decrease in ISV blood flow rates (-47%) mitigated the effects of lumen narrowing in ML1KO1 and maintained WSS levels similar to WT in the aISVs (Fig

8C) while lowering WSS levels in the vISVs (Fig 8D). Conversely, the lower reduction in blood flow rate (-33%) for ML1KO2 could not negate the induced WSS rise from lumen-narrowing and $P_{drop\_v}$ increase (only for ML1KO2 aISVs) as effectively in the aISVs (Fig 8C: 17.6%) and vISVs (Fig 8D: 16%).

In ML1OE, we saw a general maintenance of $P_{drop\_v}$ levels (Fig 8Ai and 8Bi) for both aISVs and vISVs compared to WT. ML1OE had 21% and 5.1% reductions in vessel impedance for aISVs and vISVS, respectively, compared to WT network (Fig 8Aii and 8Bii). The lowered impedance under similar $P_{drop\_v}$ conditions promoted higher blood and RBC flow rates compared to WT levels for both aISVs and vISVs (Fig 8Aiii: aISV blood flow +16%; Aiv: aISV RBC flow + 59%; Biii: vISV blood flow +11%, Biv: vISV RBC flow + 55%). Due to the mutually competing effects of vessel dilation and flow increase on WSS, the average WSS levels for aISVs in the ML1OE was statistically similar to WT (Fig 8C) while the vISV levels were slightly increased (Fig 8D, 19%).

In summary, the pressure redistribution effect of lumen size alteration has a tendency to complicate the analysis of global and local alterations to vessel lumen size in the network. Without a CFD analysis it would be difficult to predict the outcome just based on simple assumptions that vessel diameter reductions always lead to local WSS increase or vice-versa.

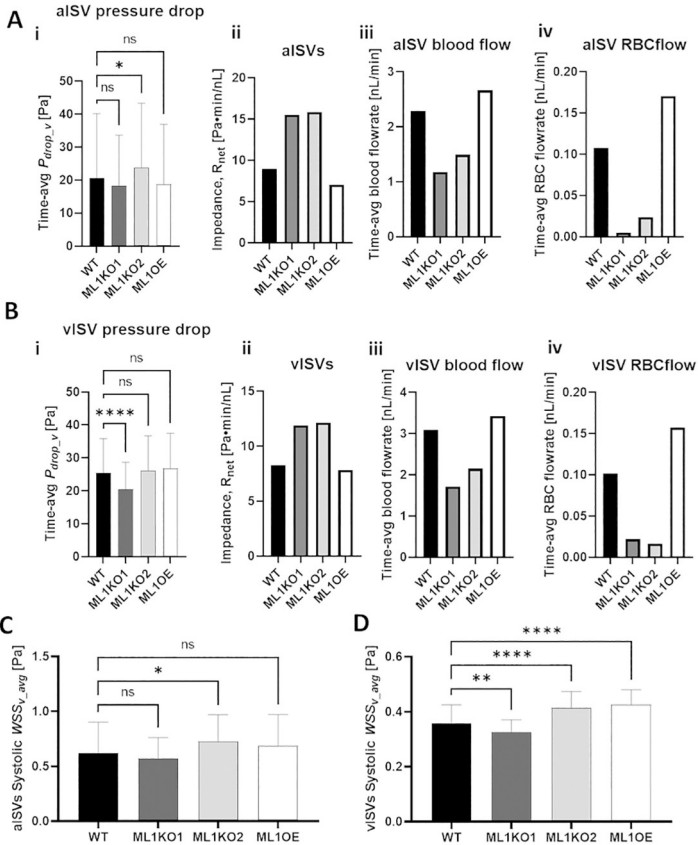

**Fig 8. Comparison of pressure and WSS distributions in ISVs amongst WT and Marcksl1 network models. (A)** Group-average (5 aISVs) of the time-averaged pressure drop ($P_{drop\_v}$) in aISVs amongst models (i) and the corresponding impedance ($R_{net}$) (ii), blood flow rate (iii) and RBC flow rate (iv) in the aISVs. **(B)** Group-average (5 vISVs) of the time-averaged pressure drop ($P_{drop\_v}$) in vISVs amongst models (i) and the corresponding $R_{net}$ (ii), blood flow rate (iii) and RBC flow rate (iv) in the vISVs. **(C)** Group-averaged (5 aISVs) systolic peak WSS ($WSS_{v\_avg}$) in aISVs amongst models. **(D)** Group-averaged $WSS_{v\_avg}$ in vISVs amongst models. Statistical comparisons were performed using one-way ANOVA with Sidak's multiple pair comparisons. *, $p < 0.05$; **, $p < 0.01$; ****, $p < 0.0001$.

More importantly, we have demonstrated that depending on the systemic adaptation to the lumen alteration, the WSS distribution transformation may either be mitigated by network pressure reduction or passively increase if network pressure is maintained. Since the experimental data has not resolved which scenario takes precedence, both scenarios could be occurring on a case by case basis between embryos.

Movies of the simulation results for RBC flow, blood pressure and WSS spatiotemporal maps for Marcksl1 OE and KO networks can be found in S7–S10 Movies.

**ISV-DLAV network mispatterning alters network hematocrit partitioning.**   The PlexinD1-Semaphorin signaling pathway controls the patterning of ISVs by repelling ECs from migrating into the somites during sprouting angiogenesis [44,45]. Here we represent a misregulation of ISV patterning based on the *plexinD1* (PlxD1) mutant phenotype with a modification of our WT geometry with 4 arterial venous shunts (AVS) bypassing the DLAV in the ISV network flow. The AVS are indicated by dashed lines in Fig 9A. Plotting the RBC trajectories and flow velocity distribution in Fig 9A (S11 Movie) indicated the tendency of RBCs to travel along the AVS paths in our PlxnD1 network model. As a result of the flow bypass and setting up of dead-flow zones in the ISV-DLAV network, only 16.4% of RBCs flowing into the ISV network enter the DLAV (Fig 9B), therefore highlighting a considerable loss of RBC perfusion into the DLAV directly due to blood shunting by the AVS.

There was a general rise in the blood and RBC flow rates in the PlxnD1 network due to the increased network connections of the plexus-like mispatterning. AVS 2 and AVS 3 were particularly prolific in increasing the RBC perfusion states (Fig 9C) of aISV2 (+193%), aISV3 (+-86.6%), vISV1 (+161%) and vISV4 (+507%). Next, we examined the WSS alterations arising from the mispatterning. From the graphical plot of systolic WSS distribution between WT and PlxnD1 networks, it can be seen that blood shunting in the PlxnD1 promoted WSS heterogeneity by accentuating both regions of high WSS around shunted vessel pathways and low WSS (indicated by red bold lines in Fig 9D, S11 Movie) around dead-flow regions. The WSS level in the PlxnD1 ISV network was higher than the WT model for both the ventral (Fig 9Ei: +43%,) and dorsal (Fig 9Eii: +25%) regions of aISVs due to the increased network flow rates. Likewise, the WSS levels in the vISVs were higher (+29%) in the PlxnD1 than the WT model for ventral locations (Fig 9Eiii) due to the increased flow from the AVS. However, the WSS in dorsal regions of vISVs for the PlxnD1 model were lower (-25%) compared to WT (Fig 9Eiv) as flow from the DLAV was heavily reduced in the phenotype.

In summary, the mispatterned network phenotype that entails blood shunting via AVS may raise the perfusion and WSS level in the modified ISV network. However, it comes at the expense of compromised perfusion to more dorsal regions of the network such as the DLAV. The resulting flow bypass instigates both perfusion and WSS heterogeneity in a dorsal versus ventral stratification of levels beyond that seen in WT.

## Discussion

In this work, we have used CFD to compare microhemodynamics in zebrafish microvascular networks of different morphologies and patterns. This was achieved by explicitly modelling the motion and deformation of RBCs in the blood mixture of cells and plasma.

The CFD model described in this work is not the first to implement cell-and-plasma flow models in examining microvascular flows. HARVEY [46], Palabos [47] and HemeLB [48] are Lattice Boltzmann Method (LBM) fluid transport solvers like our model. They have been coupled to cellular deformation models such as Hemocell [49] to model RBC deformability in cell-force spectroscopy tests [50]; WSS augmentation in capillary ECs by circulating tumor cell interactions [51]; diabetic RBCs in microaneurysm flows in retinal microvessels [52]; cellular

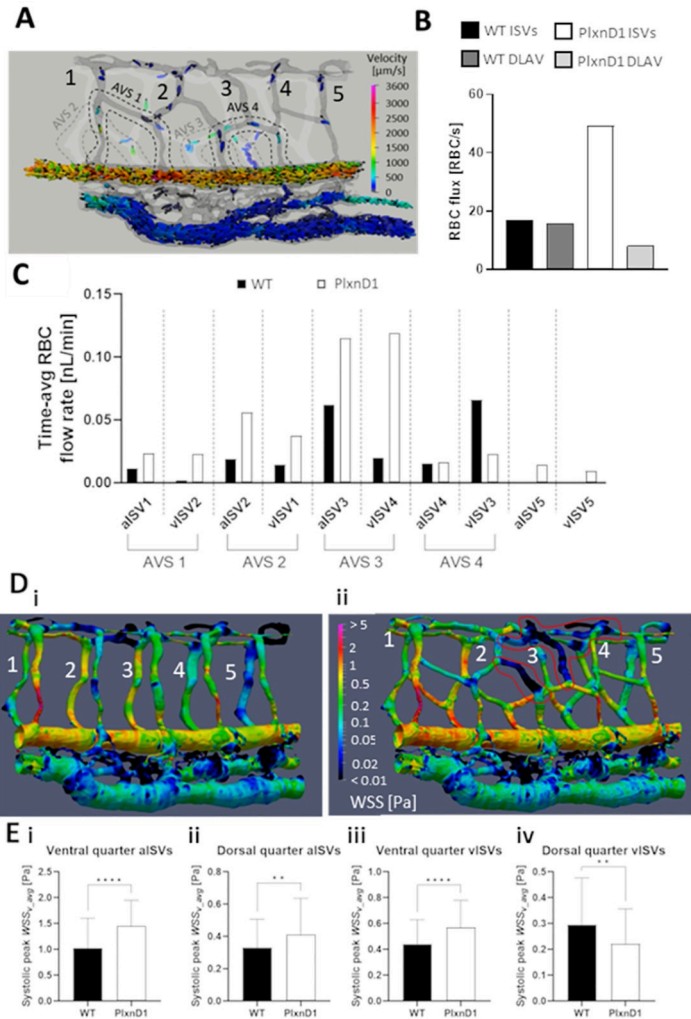

**Fig 9. Hemodynamic alterations arising from network mispatterning in PlxnD1 model.** (see S11 Movie). **(A)** Spatial distribution map of RBC trajectories/velocities at systolic peak. Dashed lines indicate arterial-venous shunts that bypass DLAV connections in the ISV network. **(B)** The flux bypass in ISV-DLAV network in PlxnD1 versus the ISV-DLAV network in WT. **(C)** Comparison of the time-averaged RBC flow rate in ISVs between WT and PlxnD1 models. **(D)** WSS spatial distribution map in WT (Di) and PlxnD1 (Dii) network models. Red bold lines in Dii indicate low-WSS dead flow regions. **(E)** Comparison of the ventral (i) and dorsal (ii) levels of systolic peak WSS between WT and PlxnD1 aISVs, the ventral (iii) and dorsal (iv) levels of systolic peak WSS between WT and PlxnD1 vISVs. Statistical comparisons in E and F were performed using unpaired two-tailed $t$ test. **, $p < 0.01$; ****, $p < 0.0001$.

margination [53], hematocrit partitioning [54], and phase-separation in RBC-rich microvascular bifurcations [28]. Likewise, other CFD solvers have studied RBC deformation and interactions with capillary walls [55], cell-free layer recovery post flow disturbances [34,56] with implications of these effects to flow and vascular pathophysiology [32]. The common impetus in all of these studies is that the RBC phase is directly responsible for the flow physiology being studied. On the other hand, some CFD studies employing experiment-derived images such as retinal microvascular beds [56] and zebrafish heart chambers [25] for WSS prediction have often omitted the explicit modelling of RBCs. By this omission, a key proponent of flow and viscosity modulation in the modelled microvascular networks is absent. Indeed, RBCs can be integral to vascular remodeling as demonstrated by Zhou and colleagues [28] where a

significantly decreased rate of CVP width reduction in *gata1* MO zebrafish was observed in comparison to control MO between 50 to 72 hpf. This implied a delayed CVP remodeling for *gata1* MO zebrafish. Our own experiments with *gata1* MO versus control MO zebrafish indicated a morphological adaptation of reduced ISV lumen diameters in the *gata1* MO group at 53 to 57 hpf. Since ISV lumen diameters have been reported to increase in tandem with rises in CA and CV hematocrit from 48 to 72 hpf [17], the ISV diameter reduction in *gata1* MO zebrafish in our study indicates a similar delay in the remodeling stage for ISVs in hematocrit-reduced zebrafish networks. Hence, our study contributes to the growing understanding that RBCs in zebrafish trunk network flow appear to participate in the remodeling of network vessels.

In terms of modelling parameters and their physiological significance, special attention has been paid to the boundary conditions (BCs) employed in our CFD model. Our basic approach to modeling is to recapitulate experiment trends in our model predictions via prescription of the appropriate pressure BCs. We found that parametric prescriptions of different pressure BCs can be treated as representations of the myriad systemic flow and pressure adaptations to hemorheological and morphological alterations observed in our zebrafish experiments. In the Marcksl1 KO analyses, 3 CFD models ML1KO1, ML1KO2 and ML1KO3 of reduced lumen diameters [43] were performed at different pressure BC levels in order to find the most appropriate flow distribution prediction that matches our experimentally measured flow trends in Marcksl1 KO zebrafish. The combined experiment-and-modeling analysis showed that the systemic reduction of lumen diameters in mutants does not elicit simple stress adaptations where lumen WSS always rise. We could recapitulate the CA flow reduction trend from WT to Marcksl1 KO networks (observed in experiments) in CFD models under two BC scenarios for the mutant models–BCs prescribing arterial pressure maintenance (from WT) in model ML1KO1 and BCs prescribing arterial pressure reduction in model ML1KO2 (Fig 7C, 7E and 7F). These two scenarios however had diametrically opposite effects on the WSS distribution, whereby pressure reduction tended to maintain WSS levels seen in the WT ISV networks and pressure maintenance tended to increase the WSS above WT ISV network levels.

Our comparison of RBC-flow model versus no-RBC flow models (represented by *gata1* morphants) highlight the possible response of a microvascular flow to a systemic reduction in blood viscosity. Supplementing the widely accepted mechanism of WSS reduction due to the lowered viscosity for heavily perfused vessels [28,39], we found that in the case of low-perfusion vessels the reduction in WSS for ISVs in *gata1* morphants was primarily due to pressure gradient weakening across the trunk network. We were able to make this inference based on the observation that CA flow rates were maintained between *gata1* and control MO groups in the experiments, regardless of the level of hematocrit reduction. By incorporating this flow maintenance adaptation trend as input into BCs in the CFD models NoRBC1 and NoRBC2, we predicted network-level weakening of pressure gradients and flow reduction into ISVs which in turn reduced ISV WSS levels.

We have additionally demonstrated the usefulness of CFD in elucidating parameters that would be difficult to control and manipulate in an experimental setting. In the SGM analyses, we highlighted the importance of vessel diameter variation along the vessel axis and the cross-sectional profile skewness in defining the vessel and network impedance. Dorsal-to-ventral narrowing of ISVs in WT vessels limits RBC entry into the network by presenting a constriction barrier and also provides the geometry for WSS gradient patterning along the vessel axis, which may serve as an essential mechanical cue for endothelial cell migration [40] and rearrangement during angiogenesis [41,42]. While both SGM1 and SGM2 indicate statistically similar levels of WSS, fine-scale differences exist between the models in how WSS varies along the circumference of the skewed lumen cross-sections in SGM1. On the other hand, WSS was

relatively homogeneous in the circumferential direction for SGM2's circular lumen cross-sections (cross-sectional WSS plots in Fig 5Bi). This may be important for studying WSS distribution and its effects on endothelial cells (ECs) lining vessel lumens. We know from monolayer cultures that ECs exposed to cell-scale differences in WSS tend to reorganize in a manner that eventually minimizes peak WSS levels and gradients [57]. Finite element analysis has shown that the heterogenous WSS distribution caused by EC topology combined with material heterogeneity around cell nucleus and focal adhesions all induce local sub-cell pockets of high strain and differentiated mechanical stimuli levels [58]. In zebrafish ISVs, ECs elongate along the lumen axis and wrap around the circumference in a unicellular lumen segment, or partial circumference in a multicellular lumen segment [59]. Consequently, regions of the same EC can be exposed to heterogenous levels of WSS depending on the local ISV topology. In such scenarios, it would be beneficial to utilize CFD models preserving all features of lumen topology to study if ECs and their sub-cellular components react to these fine-grained WSS gradients.

Another hypothetical geometry we analyzed with CFD was the PlxnD1 model of trunk ISV mispatterning. The analysis showed that shunting between aISVs and vISVs in PlxnD1 deficient networks promote flow and WSS heterogeneity in the PlxnD1 network when compared against the WT reference. This finding provides some conceptual insight into plexus hemodynamics and suggests that increased network connectivity does not always improve flow distribution as shunts often cause dead-flow regions of low RBC flow and WSS. Since WSS is known to promote vascular endothelial growth factor (VEGF) expression in ECs [60], persistent dead-flow and low WSS regions in a lumen network are expected to receive dysfunctional remodeling cues. It is known from other studies that flow and WSS participate in plexus remodeling. In a study examining DLAV remodeling, circulatory flow and VEGF expression was found to promote initial plexus expansion between 32–48 hpf and by 72 hpf promote DLAV fusion into a continuous vessel [12]. Abrogation of both flow and VEGF signaling caused regression in the DLAV remodeling. In the CVP, low WSS regions correlated with sites for intussusception whereby existing lumen split to form new plexus branches and optimize network WSS to more homogeneous distribution [10]. In avian capillary plexus remodeling, vessel sprouting occurs in low WSS locations from vessels of lower pressure towards vessels with higher pressure, often avoiding flow-convergent bifurcations [61]. In the case of mouse retina vascular plexus (postnatal day 5), local regions of stenosis and vascular regression correlated with regions of low RBC perfusion [28]. Unfortunately, the experimental studies with PlxnD1 morphants [45] and *out of bounds* (OBD) mutants [44] do not discuss the subsequent remodeling outcome of these mispatterned ISV networks. However, we can speculate from these other plexus remodeling studies that the heterogenous WSS patterning is likely to result in regression or stenosis in the low WSS dorsal ISV-DLAV connections bypassed by the shunts.

Another usefulness of CFD modeling is blood pressure prediction. The exact lumen blood pressure *in situ* is notoriously difficult to obtain from biological samples. While invasive techniques for *in situ* pressure measurements do exist, the small size of zebrafish embryos present challenges for such applications. Based on careful cross validation of other hemodynamic parameters such as RBC flow velocity, hematocrit, heart rate and even vessel wall motions in a cardiac cycle, CFD techniques could reasonably provide spatiotemporal maps of pressure in addition to WSS. These two forces together may provide a more holistic discussion of mechanotransduction and signaling cues for angiogenesis and vessel remodeling in the microvascular network.

While our study increments modeling methods and the understanding of zebrafish flow physiology and morphogenesis, limitations in scope and accuracy remain. One issue is with

the representation of flow physics at the DLAV input faces. As DLAV segments serve to complete the aISV and vISV circulatory loops, blood flow follows local lumen connections that may not follow a strict anterior to posterior directional template [12]. Our model overlooks this robust representation of DLAV flow dynamics, particularly at the 1st and 5th ISV units in the model where DLAV pressures enforced by BCs rather than local network connections dictate the flow direction. While we expect this to incur error in levels of local perfusion to specific DLAV segments, we do not expect this to change the general observations of how flow and WSS trends shift in accordance with the hemorheological and morphological alterations applied to experiments and CFD models in our study.

Another limitation is the use of group-average pooled experiment data for flow validation. As shown in Tables E and F of S1 Text (see also supplementary Figs A–E of S2 Text), each pooled group exhibits variation in the peak velocity levels and vessel diameters (Figs 3A–3F and 7B) measured. Admittedly, these biological variations make the utilization of group-averaged flow data difficult to match unique morphometric profiles of each individual network where the discrepancy between the models and experiment group average can be as high as 50% (see Table F of S1 Text). This was most apparent in the ISVs and CVs, where the experimental data saw high variance in flow velocities and RBC flux. With regards to the method of obtaining the pressure settings reported in Table B of S1 Text, a major limitation arose from not employing any optimization algorithm to iteratively correct the pressure input as we do not have this modeling expertise. We acknowledge that our approach may not rigorously demonstrate uniqueness of the pressure field distributions but we should not expect pooled-flow data to uniquely match network geometry specific to a representative zebrafish, especially in vessels that do not display strict stereotypical anatomy such as the CVP and DLAV at 2dpf.

The final limitation is that we did not discuss variations in DLAV structure and the aISV-vISV distribution pattern in the network due to the overexpansive scope of such a study. The WT CFD model reconstructed from imaging has an ISV patterning where all contralateral ISV neighbors were arterio-venous (A-V) pairings. Of the 8 ipsilateral pairings, 4 were A-V, 2 were A-A and 2 were V-V. This corresponds with findings by Geudens and colleagues who reported frequent A-V ISV pairings, especially contralaterally [7]. Since our CFD network phenotypes were direct modifications of the WT network, all phenotypes followed the same ISV pair patterning. We did not study if there were shifts in this pairing bias amongst the morphologically and hemorheologically altered zebrafish. Likewise, different DLAV architecture was not investigated as this has not been reported in Marcksl1 mutants [43] nor observed in our imaging experiments. However, on an entire trunk network scale, there may be interesting cases of local DLAV variations amongst zebrafish that is related to the pairing pattern for ISVs. In addition to the work by Geudens et al. [7], the analyses of ISV patterning and DLAV remodeling with respect to WSS patterning in the network would be enlightening. We intend to study these effects more thoroughly in a separate study.

Previous research has shown that regulation of local vascular morphology may be in synergistic feedback with network-level hemodynamics in order to optimize uniform network blood flow [62]. Occlusive effects of heavy RBC flow or clot formation in a local vessel can redistribute flow in the network around that vessel [63]. Our future plan therefore, is to study how RBC distribution asymmetries captured in both experiment and CFD models affect WSS patterning and correlate to vessel remodeling on an entire zebrafish trunk network level. To achieve this, the CFD method must be sped up. 20 cardiac pulsation cycles was our typical simulation time-scale range which allowed for RBC and plasma flow to reach developed flow conditions in our present network that is 1/6th the size of a full trunk network. Even on this reduced network, 3 weeks of simulation downtime per case was required. This downtime can be significantly reduced with further coarse-graining of the RBC deformation model as

demonstrated by Fedosov and colleagues where a low-dimensional RBC model (LD-RBC) with just 10 vertices (our RBC model has 504 vertices) could still represent the rheology of RBC-rich blood flows [20]. Furthermore, we plan to include vascular deformation responses to the oscillating pressures in the network to study vessel compliance and network capacitance trends in vascular remodeling. Lastly, for a better focus on vascular remodeling, we want to follow the developmental trajectory of the same zebrafish embryo/larvae across the first week. For this experiment and model design, we want to apply BCs and validate CFD results optimized against flow-data measured from the same fish instead of using pooled-average flow data.

## Materials and methods

### Ethics statement

All animal experiments were approved (approval no. K16-005) by the Institutional Animal Care and Use Committee at RIKEN Kobe Branch (IACUC).

### Zebrafish handling and experiments

Zebrafish (*Danio rerio*) were raised and staged according to established protocols [64]. Zebrafish lines used were *Tg(kdrl:EGFP)$^{s843}$* [65], *Tg(gata1:dsRed)$^{sd2}$* [66] and *marcksl1a$^{rk23}$; marcksl1b$^{rk24}$* [43]. For microangiography experiments, 1 nL dextran tetramethyl rhodamine (MW = 2000 kDa, Invitrogen) at 10mg/mL was injected into the Duct of Cuvier of 2 dpf zebrafish. Depletion of RBC formation was achieved by injecting 1 nL of 0.1mM *gata1* morpholino into 1- to 2-cell stage embryos.

For imaging experiments, embryos were embedded in 0.8% low-melting agarose in E3 medium containing 0.16 mg/mL Tricaine. To image RBC flow, 1000 frames at 100 fps were captured using a sCMOS camera (PCO, pco.edge 4.2 CL) mounted on a florescent stereomicroscope (Leica, M205FA). Images were processed using Fiji (NIH).

### Generation of 3D *in silico* lumen network

Lumen morphology was visualized by fluorescent dextran (introduced into the blood vessels through microangiography) and imaged using a 40X 1.25NA objective lens on an inverted Olympus IX83/CSU-Wi spinning disc confocal microscope (Yokogawa) with a Zyla 4.2 CMOS camera (Andor). Confocal z-stacks were acquired at 0.29 µm intervals and the xy pixel resolution was 0.325 µm. 3D reconstruction of lumen morphology and network was performed using Paraview (https://www.paraview.org/) to obtain an *in-silico* model of the zebrafish trunk network (S12 Movie). The 3D tif image was imported into Paraview and using the in-built software filters of "clip" and "threshold", the lumen geometry was segmented in a regionally dependent thresholding of the dextran tetramethyl rhodamine channel (channel 2). The regional customization of clipping threshold intensity was manually determined within the Paraview graphical user interface (GUI) where the EC cytosol marker channel (channel 1) was used to identify the appropriate channel 2 threshold intensities based on the boundaries where channel 1 and channel 2 voxels co-localize. This requires a graphical processing environment where 3D interrogation and manipulation of voxel data was user-friendly, which Paraview's GUI provides. Finally, the surface mesh of the lumen wall was obtained using the "Isosurface" filter (also known as "Contour" on some versions) in Paraview. For details of this procedure please refer to S12 Movie.

## Computational fluid dynamics (CFD) model

The three major solver components to our blood flow model are **1)** the coarse-grained spectrin model (CGSM) for predicting the red blood cell (RBC) deformation by flow forces, **2)** the lattice Boltzmann method (LBM) for pressure and velocity update of the plasma and cytoplasmic fluid and **3)** the immersed boundary method (IBM) for calculating the exchange of momentum between the RBC membrane and the surrounding fluid.

**RBC deformation model: Coarse-grained spectrin model (CGSM).** The zebrafish RBC is modelled as a flattened ellipsoid capsule (10 by 7 μm in the long and short disc axes, respectively and 3 μm in thickness [67]) with an RBC membrane shell enveloping a cytosolic interior and a nucleus (Fig 1C). The membrane is represented by a hexagonal mesh where edges represent the cytoskeleton (CSK) of spectrin chains joined to actin filaments [68,69] and the triangle surface elements represent the plasma membrane (PM). A major difference between zebrafish RBCs and human RBCs apart from the shape, is the presence of nuclei even in mature RBCs for zebrafish [4,70]. Our model therefore includes the nucleus as a three-dimensional meshwork where enjoining edges act as compressible springs for the prescription of nuclear deformation. Using a total of 504 membrane points per RBC, the CGSM enforces the shearing resistance ($U_{spring}$) [71] conferred by the CSK (Fig 10Ai), bending ($U_{bend}$) [72] and areal-expansion ($U_{area}$) [73] resistance conferred by the PM (Fig 10Aii) and volumetric expansion resistance ($U_{volume}$) conferred by an incompressible cytosolic interior. Deformation forces ($\boldsymbol{F}_{m,i}$) in the RBC membrane point $\boldsymbol{i}$ (Fig 10Bi) may be obtained by differentiating the local elastic energy ($U_{elastic,i}$) with respect to the nodal displacement ($\boldsymbol{s_i}$) produced by the deformation:

$$\boldsymbol{F}_{m,i} = \frac{\partial U_{elastic,i}}{\partial \boldsymbol{s_i}} \ where \ U_{elastic} = U_{spring} + U_{bend} + U_{area} + U_{volume} \tag{4}$$

For more details into the constitutive laws employed by the CGSM, please refer to section F of S2 Text. For validation of the material deformation properties of RBC membranes using the CGSM please refer to Supplementary material Section C of [36].

**Plasma and cytosol transport model: Lattice Boltzmann Method (LBM).** The LBM discretizes the mass and momentum of fluid particles (microstates) using the density distribution function ($f_i(\boldsymbol{x})$) on a fixed computational lattice. Microstates were updated in a two-stage solver process by streaming and collision of $f_i(\boldsymbol{x})$ at every time step using the Bhatnagar Gross and Krook (BGK) relaxation model. The D3Q19 fluxing stencil was used for streaming in the 19 lattice velocity directions ($\boldsymbol{c_i}$) [74]. In the collision step of the LBM, momentum imparted to the plasma from RBC deformation forces was represented using the body acceleration term $\boldsymbol{B}_i$ added to the LBM-BGK model [75]:

$$f_i(\boldsymbol{x} + \boldsymbol{c_i}\Delta t, t + \Delta t) - f_i(\boldsymbol{x}, t) = -\frac{|f_i(\boldsymbol{x}, t) - f_i^{eq}(\boldsymbol{x}, t)|}{\tau} + \Delta t \boldsymbol{B}_i \tag{5a}$$

$$where \ \tau = \frac{1}{2} + \frac{\mu}{\rho c_s^2 \Delta t} ; c_s = \frac{\Delta x}{\Delta t} / \sqrt{3} \tag{5b}$$

where $\Delta x$ is the spatial grid step size in the lattice, $\Delta t$ is the time step size, and symbol $i$ (19 directions) denotes the flux direction index. $\rho$ is the fluid density, $\mu$ is the dynamic viscosity, $c_s$ is the speed of sound in the LBM and $\tau$ is the BGK relaxation parameter towards the

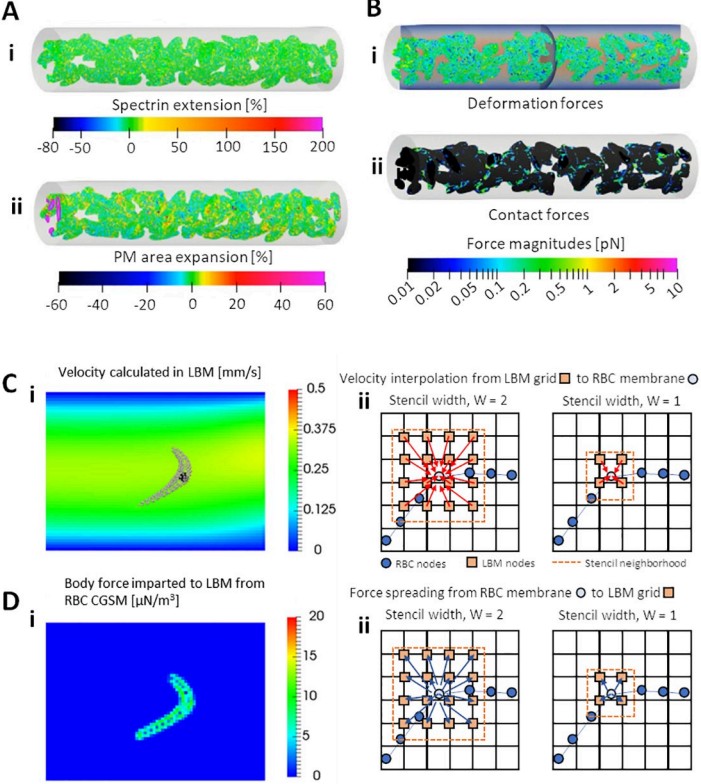

**Fig 10. The fundamental components and physics of the cell-and-plasma phase CFD model. (A)** Membrane deformation characteristics modelled by the coarse-grained spectrin model (CGSM) in terms of spectrin extension (i) and plasma membrane areal expansion (ii). **(B)** Forces arising in the CGSM due to membrane deformations (i) and cell-cell & cell-lumen-wall contact (ii). **(C)** Schematics of the velocity update in the immersed boundary method (IBM) where surrounding fluid velocities calculated by the lattice Boltzmann method (LBM) solver (i) are interpolated to membrane point in accordance with Eqs (7) & (8) and the stencil width $W$ (ii). **(D)** Schematics of the momentum update in the immersed boundary method where forces of deformation in the membrane (solved by CGSM) are spread to the fluid as body-forces (i) in accordance with Eqs (8) & (9) and the stencil width $W$ (ii).

equilibrium distribution ($f_i^{eq}(\boldsymbol{x}, t)$) which can be obtained through:

$$f_i^{eq}(\boldsymbol{x}, t) = w_i\, \rho \left[ 1 + \frac{\boldsymbol{c_i} \cdot \boldsymbol{u}}{c_s^{\,2}} + \frac{1}{2}\frac{(\boldsymbol{c_i} \cdot \boldsymbol{u})^2}{c_s^{\,4}} - \frac{1}{2}\frac{|\boldsymbol{u}|^2}{c_s^{\,2}} \right] \tag{6}$$

where $\boldsymbol{u}$ is the velocity; $\rho$ and $\boldsymbol{u}$ are calculated by: $\rho = \Sigma_i f_i$, $\boldsymbol{u} = \Sigma_i f_i\, \boldsymbol{c_i}/\rho$. The weights, $w_i$, used in Eq (6) are given by $w_0 = \frac{1}{3}$, $w_i = \frac{1}{18}$ for $i = 1 - 6$ and $w_i = \frac{1}{36}$ for $i = 7 - 18$.

The body acceleration term in Eq (5a) is given by:

$$\boldsymbol{B}_i = \left( 1 - \frac{1}{2\tau} \right) \omega_i \left[ \frac{\boldsymbol{c_i} - \boldsymbol{u}}{c_s} + \frac{\boldsymbol{c_i} \cdot \boldsymbol{u}}{c_s}\boldsymbol{c_i} \right] \cdot \boldsymbol{F}_f \tag{7}$$

where $\boldsymbol{F}_f$ is the body force passed to the fluid from the fluid-structure interaction with the CGSM (RBC solver).

**Fluid structure interaction: Immersed boundary method (IBM).**    The IBM introduced by [76] enforces the fluid-structure interaction in the simulation by providing the bidirectional coupling between the fluid motion and the membrane dynamics. Velocity interpolations onto

the membrane surface grid from the LBM-derived velocity field (Fig 10Ai) using the discrete delta function were performed to update the velocity vectors ($\boldsymbol{u}_m$) and position vectors ($\boldsymbol{x}_m$) of the RBC membrane nodes as follows:

$$\boldsymbol{u}_m(\boldsymbol{x}_m) \sum_f \delta(\boldsymbol{x}_f - \boldsymbol{x}_m) \boldsymbol{u}(\boldsymbol{x}_f) \tag{8}$$

where $\delta(\boldsymbol{x}_f - \boldsymbol{x}_m)$ is the discrete delta function that is given by:

$$\delta(\boldsymbol{x}_f - \boldsymbol{x}_m) = \begin{cases} 0 & |x_{i,f} - x_{i,m}| > W\Delta x \\ \prod_{i \in (1,2,3)} \frac{1}{2W} \left( 1 + \cos \frac{\pi(x_{i,f} - x_{i,m})}{W\Delta x} \right) & otherwise \end{cases} \tag{9}$$

where $\boldsymbol{x}_f$ is the position vector of the surrounding fluid lattice grid points, $x_i = \{x_1, x_2, x_2\} = x$, $y$, $z$ and $W$ is the stencil width for the delta function in terms of $\Delta x$ (Fig 10Cii and 10Dii).

The viscoelastic resistance of the RBC membrane against hydrodynamic forces and its effect on the surrounding fluid is similarly enforced whereby the membrane force $\boldsymbol{F}_m(\boldsymbol{x}_m)$ arising from the membrane deformation at membrane coordinates $\boldsymbol{x}_m$ is distributed to the surrounding fluid nodes (coordinates $\boldsymbol{x}_f$) (Fig 10Bi and 10Bii) as a body force $\boldsymbol{F}_f(\boldsymbol{x}_f)$ using the discrete delta function:

$$\boldsymbol{F}_f(\boldsymbol{x}_f) = \frac{1}{\Delta x^3} \sum_m \delta(\boldsymbol{x}_f - \boldsymbol{x}_m) \boldsymbol{F}_m(\boldsymbol{x}_m) \tag{10}$$

**RBC contact mechanics and solver time-step staggering.** In the LBM, $\Delta t$ is constrained by the relaxation parameter $\tau$ which needs to be kept close to 1 in order to minimize numerical diffusion at wall boundary conditions (Fig I of S2 Text). On the other hand, the CGSM is afforded numerical stability by well-tested scaling laws [77] that allow larger time steps than the LBM. Hence, we saved computational time by coarsening the calculation and update frequencies of the CGSM solver and IBM solver to one CSGM and IBM iteration at every 50 LBM intervals ($\Delta t_{CGSM} = 50\Delta t_{LBM}$). Consequently, the staggered time management between solvers required explicit checking for contact conditions from RBC to RBC, RBC to cell nucleus and RBC to lumen wall at each CGSM/IBM update interval. For the RBC to RBC contact, we looped the data list for local neighboring surface points ($s$) around an RBC membrane node ($m$) at every $\Delta t_{CGSM}$ to check for inter-surface separation distances below the separation threshold condition ($r_{sep} < r_{sep}^*$). For surfaces below this threshold, the inter-RBC surface repulsion velocity ($\boldsymbol{u}_{m\_RBCrepul}$) was updated accordingly:

$$\boldsymbol{u}_{m\_RBCrepul} = \sum_s u_{repul} \left( \frac{r_{sep}^* - r_{sep}}{r_{sep}^*} \right) \hat{e}_s \quad when \quad r_{sep} < r_{sep}^* \tag{11a}$$

where $u_{repul}$ is the coefficient for bounce-back velocity, $r_{sep}$ and $r_{sep}^*$ are the inter surface separation distance and separation threshold respectively. $\hat{e}_s$ is the unit vector pointing from the neighbor surface point back to the membrane node:

$$r_{sep} = |\boldsymbol{x}_m - \boldsymbol{x}_s|; \hat{e}_s = \frac{\boldsymbol{x}_m - \boldsymbol{x}_s}{r_{sep}} \tag{11b}$$

where $\boldsymbol{x}_s$ is the position vector of the neighboring RBC surface point.

Likewise, repulsion between an RBC membrane node ($m$) and the neighboring nucleus nodes ($n$) in the cell interior was enforced by looping through the data lists for nearby $n$ and sub-triangle membrane mesh elements (*tri*) directly adjacent to $m$. When the separation distance between *tri* and $n$ were below the $r_{sep}{}^*$ threshold, repulsion velocity from nucleus nodes ($\boldsymbol{u}_{m\_Nucrepul}$) to $m$ was incremented as follows:

$$\boldsymbol{u}_{m\_Nucrepul} = \frac{1}{3} \sum_{t,n} u_{repul} \left( \frac{r_{sep}{}^* - r_{sep}}{r_{sep}{}^*} \right) \hat{e}_{tri} \quad when \quad r_{sep} < r_{sep}{}^* \tag{12a}$$

$$where \; r_{sep} = |\boldsymbol{x}_{tri} - \boldsymbol{x}_n| \tag{12b}$$

$\hat{e}_{tri}$ is the normal vector on the RBC sub-triangle mesh element (*tri*) around the membrane node ($m$), $\boldsymbol{x}_{tri}$ and $\boldsymbol{x}_n$ are the position vectors of *tri* and nucleus node ($n$).

Repulsion between an RBC membrane node ($m$) and the closest lumen wall mesh surface point ($w$) was enforced when the separating distances were below the $r_{sep}{}^*$ threshold:

$$\boldsymbol{u}_{m\_Wallrepul} = \begin{cases} u_{repul} \left( \dfrac{r_{sep}{}^* - r_{sep}}{r_{sep}{}^* + 0.75} \right) \hat{e}_w & when \quad -0.75 \leq r_{sep} < r_{sep}{}^* \\[2mm] u_{repul} \hat{e}_w & when \quad r_{sep} < -0.75 \\[2mm] 0 & when \quad r_{sep} \geq r_{sep}{}^* \end{cases} \tag{13a}$$

$$where \; r_{sep} = (\boldsymbol{x}_m - \boldsymbol{x}_w) \cdot \hat{e}_w \tag{13b}$$

$\hat{e}_w$ gives the direction of the wall to RBC bounce-back, which is the normal vector on the closest lumen wall surface mesh point ($w$). Note that all distances in Eqs (11)–(13) are in µm.

As shown in Eq (13), the scenario of RBC mesh crossing the lumen wall boundaries (when $r_{sep} < 0$) is permitted so that the model does not enforce artificial cell-free plasma layer development in narrow vessels like the ISVs. While RBC-lumen-wall mesh penetration events are rare in the computation, they can occur occasionally due to the large time-step taken. To prevent prolonged scenarios of such domain exit, the magnitude of bounce-back is scaled up to ensure RBCs re-enter the lumen. Robustness of the cell-contact model was maintained using the $u_{repul}$ and $r_{sep}{}^*$ values summarized in Table A of S1 Text.

The velocity of the local membrane node $\boldsymbol{u}_m(\boldsymbol{x}_m)$ evaluated in Eq (8) was modified by addition of the inter-surface repulsion velocities to the interpolated velocity obtained from IBM:

$$\boldsymbol{u}_m(\boldsymbol{x}_m) = \left[ \sum_f \delta(\boldsymbol{x}_f - \boldsymbol{x}_m) \boldsymbol{u}(\boldsymbol{x}_f) \right] + \boldsymbol{u}_{m\_RBCrepul} + \boldsymbol{u}_{m\_Nucrepul} + \boldsymbol{u}_{m\_Wallrepul} \tag{14a}$$

$$\boldsymbol{x}_m(t + \Delta t_{CGSM}) = \boldsymbol{x}_m(t) + \boldsymbol{u}_m(\boldsymbol{x}_m) \Delta t_{CGSM} \tag{14b}$$

The explicit correction of the RBC mesh node velocity required momentum imparted back to the surrounding fluid by modifying Eq (9) with the addition of the force-correction term

$F_{contact}$ (Fig 10Bii):

$$F_f(\boldsymbol{x}_f) = \frac{1}{\Delta x^3} \sum_m \delta(\boldsymbol{x}_f - \boldsymbol{x}_m)[\boldsymbol{F}_m(x_m) + \boldsymbol{F}_{contact}] \qquad (15a)$$

$$where\ \boldsymbol{F}_{contact} = \eta_{fluid}\left(\boldsymbol{u}_{m\_RBCrepul} + \boldsymbol{u}_{m\_Nucrepul} + \boldsymbol{u}_{m\_Wallrepul}\right) \qquad (15b)$$

where $\eta_{fluid}$ represents a fluid dissipative friction to velocity corrections ($\eta_{fluid}$ = 1 x $10^{-7}$ N·s/m).

**Grid independence testing.** To ascertain the performance matrix of accuracy against computing time in our CFD method, we performed grid resolution independence on two reduced domain sets. The first set was a 40 μm long straight vessel segment that was 8 μm in diameter, which represented an ISV-type vessel test segment (Fig 11A and 11B). Applying a pressure drop of 18 Pa from left to right end, the steady state flow solution was obtained under various grid resolutions: very fine (VF) at $\Delta t_{LBM}$ = 0.125 μs, $\Delta x$ = 0.125 μm; fine (F) at $\Delta t_{LBM}$ = 0.25 μs, $\Delta x$ = 0.25 μm; medium (M) at $\Delta t_{LBM}$ = 0.5 μs, $\Delta x$ = 0.5 μm; coarse (C) at $\Delta t_{LBM}$ = 1 μs, $\Delta x$ = 1 μm.

The second set was a 150 μm long straight vessel segment that was 30 μm in diameter, which represented a CA/CV-type vessel test segment (Fig 11C and 11D). The VF resolution

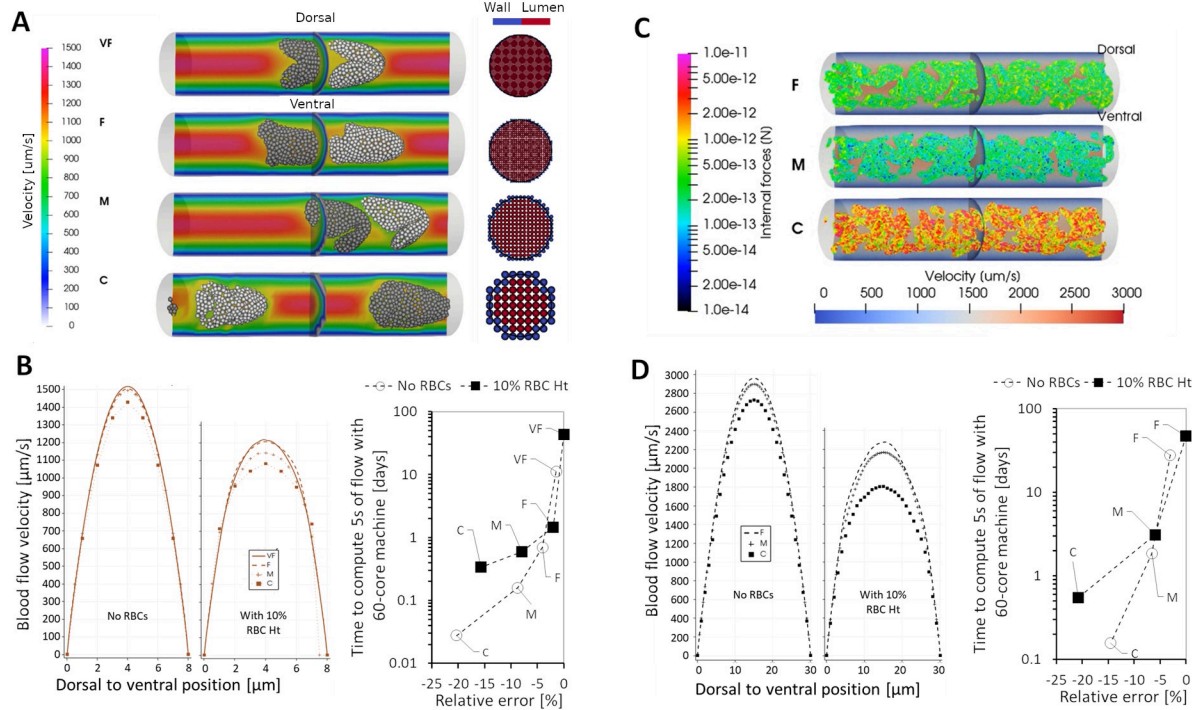

**Fig 11. Grid independence testing for numerical accuracy and precision in the simulations. (A)** Snapshots of the ISV-domain grid testing simulations with an 8 μm diameter by 40 μm length vessel segment under steady-state 18 Pa flow-driving pressure. The simulations were performed with computational grids at very fine (VF), fine (F), medium (M) and coarse (C) grid resolutions. For clarification of the grid resolution dimensions refer to main text. Plot fills are colored for flow velocity on the left image while circle cross sections on the right image indicate the grid voxels which are colored in red for lumen and blue for vessel wall. **(B)** The predicted flow velocities under varying grid resolutions (left) and the computing-cost versus simulation error matrix (right). **(C)** Snapshots of the CA-domain grid testing simulations with a 30 μm diameter by 150 μm length vessel segment under steady-state 18 Pa flow-driving pressure. The simulations were performed with computational grids at fine (F), medium (M) and coarse (C) grid resolutions. Plot fills are colored for flow velocity on the left image. **(D)** The predicted flow velocities under varying grid resolutions (left) and the computing-cost versus simulation error matrix (right).

was omitted for this larger test case. Applying a pressure drop of 9.6 Pa, the steady state flow solution was obtained under F, M and C grid resolutions.

Both sets were first performed without RBCs in the simulation and compared against the theoretical Poiseuille flow solution ($v_{peak}$ = 1500 μm/s, Q = 3.77x10$^4$ μm$^3$/s for ISV-test; $v_{peak}$ = 3000 μm/s, Q = 1.06x10$^6$ μm$^3$/s for CA/CV-test). The simulated velocity profiles indicated a general improvement in velocity representation as the grid resolution was improved (Fig 11Bi and 11Di). The flow rate (Q) errors for ISV-test compared to Poiseuille theory were -1.5% for VF, -4% for F, -8.8% for M and -20.3% for C (Fig 11Bii). Likewise, the errors for CA/CV-test were -3.1% for F, -6.6% for M and -14.5% for C (Fig 11Dii).

Next, we included RBCs in the test domains (S13 and S14 Movies) to profile the combined computing load when LBM, CGSM and IBM were all active under the time-staggering approach. Here, the stencil width *W* for the delta function in the IBM was 2 in VF and F, 1 in M and C (see Eq (8)). Using parallel computing with 60 threads, we simulated 0.5s of RBC flow in all test models. Considering that the network flows in the main study require around 5s of repeated cardiac oscillations, we set this as the target computing cycle for comparison between the grids. As such, we multiplied the time taken in the 0.5s test cycles by 10 to obtain the 5s estimate. Unlike the tests without RBCs, we do not have a theoretical solution to compare the grid independence set against. Hence, the error is given relative to the highest resolution tested: VF for the ISV test and F for the CA/CV test. Relative errors were -2.0% for F, -7.9% for M and -15.7% for C compared to VF in the ISV test (Fig 11Bii). Relative errors were -6.0% for M and -20.8% for C compared to F in the CA/CV test (Fig 11Dii). Importantly, the time taken to simulate 5s of flow in the ISV test was estimated to be 43 days for VF, 1.5 days for F, 0.6 days for M and 0.3 days for C. The time taken for 5s of flow in the CA/CV test was estimated to be 47 days for F, 3.1 days for M and 0.5 days for C.

Extrapolating the performance matrix to the zebrafish trunk network domain in the main study (5 times larger than the CA/CV test domain), we determined that a 15-day downtime for a 6–8% error tradeoff in the M grid was most feasible for our zebrafish network analysis. An F level grid on the other hand would require a 230 day computing downtime for each main study case analyzed on a 60-thread parallel computing processor (AMD Ryzen Threadripper 3970X CPU). Further grid coarsening to C level on the other hand incurs heavy accuracy issues and poor qualitative representation of RBC deformation behavior. This was apparent in the ISV test where VF, F and M all showed similar parachute folding profiles of the RBCs, unlike the ruffled folds seen in RBCs in the C grid model (Fig 11A). In the CA/CV test, the RBCs in C grid model had deformation forces 3–10 times larger than M and F grids for the same shearing rates applied, thus indicating poor resolution of membrane deformations between the CGSM and LBM via the IBM (Fig 11C). Subsequently, we employed M level grids in our main study, where $\Delta t_{LBM}$ = 0.5 μs and $\Delta x$ = 0.5 μm. We attempted to employ IBM stencil sizes of *W* = 1 for the zebrafish trunk networks in the main study but found the narrow stencil setting prone to unstable RBC-wall contact dynamics at irregular lumen wall surface geometries near vessel junctions (Fig Ji of S2 Text). Consequently, the M grids in the main study utilized a more stable IBM stencil scheme with *W* = 2 (Fig Jii of S2 Text). As a consequence of the broader stencil used in the main study, flow levels in models with *W* = 2 setting were notably underpredicted as compared to *W* = 1 setting (Fig Jiii of S2 Text). Thus, we estimated that a further 12% error (estimated accumulated error of 20% against F grid level) was incurred in the M-grids used in our main study due to an artificial viscosity augmentation effect of the IBM stencil diffusion (Fig Jiv of S2 Text).

## Supporting information

**S1 Text. Tables for simulation and experiment data.**
(PDF)

**S2 Text. Supplementary text for A cell-and-plasma numerical model reveals hemodynamic stress and flow adaptation in zebrafish microvessels after morphological alteration.**
(PDF)

**S1 Movie. Oscillating RBC flow velocities, wall shear stress (WSS) and blood pressure distribution in the WT network.**
(MP4)

**S2 Movie. Inter-cellular collisions between deformable red blood cells (RBCs) and RBC-vessel-wall collisions can produce extensive deformation of impinged RBCs and cause flow passage reductions.** Utilization of short-range repulsion forces mediated contact mechanics and ensured a robust numerical solver.
(MP4)

**S3 Movie. Simulation of RBC flow in a 30 μm diameter and 150 μm long vessel segment with steady-state 9.6 Pa flow-driving pressure and under no RBC aggregation conditions for 10% hematocrit versus 20% hematocrit.**
(MP4)

**S4 Movie. Simulation of RBC flow in a 30 μm diameter and 150 μm long vessel segment at 10% hematocrit (Ht) with steady-state 9.6 Pa flow-driving pressure and under no RBC aggregation versus 1 μJ/m$^2$ aggregation conditions.**
(MP4)

**S5 Movie. Simulation of RBC flow in a 30 μm diameter and 150 μm long vessel segment at 20% hematocrit (Ht) with steady-state 9.6 Pa flow-driving pressure and under no RBC aggregation versus 1 μJ/m$^2$ aggregation conditions.**
(MP4)

**S6 Movie. Comparison of the oscillating RBC flow velocities, pressure maps and WSS maps in the smooth geometry model (SGM) networks SGM1, SGM2 and SGM3.**
(MP4)

**S7 Movie. Oscillating RBC flow velocities, spatial distributions of blood pressure and WSS in the Marckl1 over-expression phenotype (Marcksl1 OE) network.**
(MP4)

**S8 Movie. Oscillating RBC flow velocities, spatial distributions of blood pressure and WSS in the *marckl1a;marckl1b* double knockout phenotype (Marcksl1 KO) network with 22% reduced arterial pressure compared to WT (ML1KO1).**
(MP4)

**S9 Movie. Oscillating RBC flow velocities, spatial distributions of blood pressure and WSS in the Marcksl1 KO network with same arterial pressure compared to WT (ML1KO2).**
(MP4)

**S10 Movie. Oscillating RBC flow velocities, spatial distributions of blood pressure and WSS in the Marcksl1 KO network with 17% increased arterial pressure compared to WT (ML1KO3).**
(MP4)

**S11 Movie. Oscillating RBC flow velocities, spatial distributions of blood pressure and WSS in the PlexinD1 knockout (PlxnD1) network.**
(MP4)

**S12 Movie. 3-D image segmentation process for generation of CFD mesh from two-channel confocal z-stack images with endothelial cell marker and lumen volume marker.**
(MP4)

**S13 Movie. Grid independence testing for an ISV-like 8 μm diameter by 40 μm length vessel segment: RBC deformation and blood flow velocity prediction under computational grids in descending order of spatial and temporal resolution from very fine (VF), fine (F), medium (M) to coarse (C).**
(MP4)

**S14 Movie. Grid independence testing for a CA-like 30 μm diameter by 150 μm length vessel segment: RBC deformation and blood flow velocity prediction under computational grids in descending order of spatial and temporal resolution from fine (F), medium (M) to coarse (C).**
(MP4)

## Acknowledgments

We thank members of the Phng Lab and Satoru Okuda for discussions and RIKEN BDR Aquatic Facility for zebrafish care.

## Author Contributions

**Conceptualization:** Li-Kun Phng.

**Data curation:** Swe Soe Maung Ye.

**Formal analysis:** Swe Soe Maung Ye.

**Funding acquisition:** Li-Kun Phng.

**Investigation:** Swe Soe Maung Ye.

**Methodology:** Swe Soe Maung Ye.

**Project administration:** Li-Kun Phng.

**Software:** Swe Soe Maung Ye.

**Supervision:** Li-Kun Phng.

**Writing – original draft:** Swe Soe Maung Ye.

**Writing – review & editing:** Swe Soe Maung Ye, Li-Kun Phng.

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
