## [Decision Letter · Decision Letter 0]

14 Aug 2023

Dear Dr. Phng,

Thank you very much for submitting your manuscript "A cell-and-plasma numerical model reveals hemodynamic stress and flow adaptation in zebrafish microvessels after morphological alteration" for consideration at PLOS Computational Biology.

As with all papers reviewed by the journal, your manuscript was reviewed by members of the editorial board and by several independent reviewers. In light of the reviews (below this email), we would like to invite the resubmission of a significantly-revised version that takes into account the reviewers' comments.

We cannot make any decision about publication until we have seen the revised manuscript and your response to the reviewers' comments. Your revised manuscript is also likely to be sent to reviewers for further evaluation.

Sincerely,

Alison L. Marsden

Academic Editor

PLOS Computational Biology

Jason Haugh

Section Editor

PLOS Computational Biology

Reviewer's Responses to Questions

**Comments to the Authors:**

Reviewer #1: The authors have carefully addressed the reviewers’ comments (including my own) with additional simulations and benchmark tests, and I find the quality of the paper largely improved and scientifically sound. However, the minor issues below should be addressed before its publication.

1. The revised manuscript is very dense with overflown data & analysis, and difficult to navigate. One reason behind this is the evenly detailed/weighted description of less important data (e.g. exhaustively listed percentages which are not needed everywhere or statistically insignificant values). Another reason is the frequent jumping between experiment and simulation data, sometimes without clear signposting. The authors may consider adding a signposting paragraph at the beginning of the results section to offer a point of entry before the readers dive in the comprehensive results sections.

2. Related to point 1, avoid confusing classification of cases, e.g. “Model 1, Model 2, Model 3” for the WT in section 2.1 and “model 1, model 2, model 3” again for the Marcksl1 KO in section 2.2 (consider distinguishing them in both the figures and text). Some supplementary movies may be combined, e.g. velocity, pressure and WSS videos for the same case.

3. Not sure the individual section 2.3.2 for pressure redistribution is necessary, which is linearly correlated with WSS for designated vessel segments.

4. Section 2.3.3, last paragraph: “based on vessel impedance estimated by vessel size”. This contrasts with the definition of vessel impedance on Page 9.

5. In the discussion section (e.g. page 16, paragraph 1), the authors are suggested to link findings in this study with prior studies, either supporting or contradicting current understanding of how mechanical cues shape vascular development (as nicely discussed in the introduction section). A missing opportunity is to bridge the findings of vascular morphology-triggered flow adaptation in this study with potential WSS heterogeneity-triggered vascular remodelling as reported by previous studies. Also, the RBC flow studied here should be compared with similar studies, such as the cited work (Zhou et al., 2021), who observed fairly dynamical RBC perfusion in the CVP of zebrafish vasculature and provided some insights on the role of RBCs in remodelling the vascular network.

6. Why adopt pressure boundary conditions at the CA/CV inlets/outlets? Isn’t it easier to directly prescribe inflow/outflow BCs given these are experimentally measured already? This way, designated discharged haematocrits can also be easily fed at the inlets for given flow rates.

7. The authors should briefly justify why nucleated RBCs are modelled in relation to the 2 dpf zebrafish studied (also, does the RBC size match?). Note a potential conflict of zebrafish age, 53-57 hpf on page 7.

8. In addition to Hemocell (Zavodszky et al, 2017), other widely used cell-plasma LBM flow solvers should also be mentioned, such as Hemelb (Zhou et al., 2021), Palabos (Latt et al., 2020: 10.1016/j.jocs.2020.101153) and Harvey (Ames et al., 2020: 10.1016/j.camwa.2020.03.022)

Reviewer #2: The manuscript presents a 3-D+time Computational Fluid Dynamics (CFD) model depicting microcirculation in embryonic zebrafish tail, which incorporates a blood transport model of cells and plasma and offers novel insights into microvascular circulation, remodeling, and adaptation. To further strengthen the paper and its relevance to the intended readership, the following points are suggested for revision:

Major comments:

- A noticeable gap in the manuscript pertains to the image segmentation and analysis needed to construct the 3-D computational domain. The process of image segmentation, necessary for the reconstruction of the vessel network, is currently unclear. Further explanation and details regarding the segmentation algorithm would be beneficial. The usage of Paraview for this purpose seems unorthodox; could you confirm if a thresholding algorithm was applied given the apparent non-smoothness in the computational domain (Fig.1 B – ISVs at left)?

- The flow patterns, notably in ISVs, are heavily influenced by the structure and connectivity of the DLAV. The addition of a discussion that evaluates the reliability of using the DLAV structure and connectivity pattern from a particular section of one zebrafish tail, and whether these findings can be generalized, would potentially support the argument and overall conclusions. The model, as it stands, presumes a paired aISV and vISV at each branching point along the CA, which is not always the case. This observation extends to the application of changes in vessel diameter in the Marksl1 models. The manuscript does not provide clarity on the specific section of the zebrafish tail (anterior or posterior) considered for the experimental diameter measurements.

- In Table 2, are the listed values specifically tuned to align with the experimental range? If so, it would be useful to know which algorithm was used for this purpose and how unique this particular combination of parameters is.

- The blood flow in DLAV can be towards the posterior or anterior sections. How is the DLAV pressure boundary condition decided in these faces at the two sides if the zebrafish line used for vessel segmentation only represents the vessel structure?

- Minor comments:

- Some sections of the manuscript require rephrasing for improved clarity and readability. For instance, the explanation "Flow velocities were set at the two periodic faces of the reservoir domains using the velocity at the shared boundary faces with the main network domain" could be made clearer. Furthermore, it might be advantageous to relocate the percentage changes in the comparisons to the supplementary material, enabling a focused presentation of the main outcomes in the body of the paper given the volume of the materials being presented.

- On Page 7, Line 18, could the authors elaborate on the term 'lumen-area weighted RBC count'? What is the 'area' referred to in this context?

- In Figure 3, both the control and morpholino groups incorporate 7-9 zebrafish, yet it appears that only four from the control group show a higher relative hematocrit in CA. The coloring used for the two groups is identical and might cause confusion; it would be beneficial to differentiate between the groups visually.

- For Figure 3F, can the authors specify the method used to measure the vessel diameter in the model, and the number of aISV and vISV included? The flow rate along the dorsal aorta can vary depending on the distal location as some flow diverts through the ISVs, decreasing towards the posterior tail. Similarly, the size of ISVs is dependent on the location along the tail in a developing embryonic zebrafish. What criteria were used to ensure that the vessels originated from a similar region?

- What is the rationale for considering the diameter's dependence on the hematocrit as linear while estimating the diameter change in Model 3?

- Figure 7.C indicates that the low perfusion group has an RBC flux close to zero. Given that RBC trajectories are employed for vessel diameter estimation, has a sufficient number of ISVs from the zebrafish in the low perfusion group been utilized to measure the diameter presented in section B?

- Figure 9 appears to be missing panel F.

- The authors assert that a key advantage of their model is its ability to represent local topologies. Although 3-D modeling of zebrafish tail structures is not a novel concept, a discussion on the in-plane image resolution used for the reconstruction and the number of in-plane pixels along the diameter would likely be vital and beneficial.

**Have the authors made all data and (if applicable) computational code underlying the findings in their manuscript fully available?**

Reviewer #1: Yes

Reviewer #2: Yes

PLOS authors have the option to publish the peer review history of their article (what does this mean?). If published, this will include your full peer review and any attached files.

Reviewer #1: No

Reviewer #2: No
---

## [Decision Letter · Decision Letter 1]

6 Nov 2023

Dear Dr. Phng,

We are pleased to inform you that your manuscript 'A cell-and-plasma numerical model reveals hemodynamic stress and flow adaptation in zebrafish microvessels after morphological alteration' has been provisionally accepted for publication in PLOS Computational Biology.

Best regards,

Alison Marsden

Academic Editor

PLOS Computational Biology

Jason Haugh

Section Editor

PLOS Computational Biology

Reviewer's Responses to Questions

**Comments to the Authors:**

Reviewer #1: The authors have satisfactorily addressed my comments in this revision and I hereby recommend publication of the study. However, I would remind the authors that the experimental element of their work is insufficiently reflected in the current abstract, which they may want to improve before publication.

Reviewer #2: The authors have addressed the revisions and concerns, and we recommend the acceptance of this manuscript for publication.

**Have the authors made all data and (if applicable) computational code underlying the findings in their manuscript fully available?**

Reviewer #1: Yes

Reviewer #2: Yes

PLOS authors have the option to publish the peer review history of their article (what does this mean?). If published, this will include your full peer review and any attached files.

Reviewer #1: No

Reviewer #2: No

---

## [Editor Report · Acceptance letter]

29 Nov 2023

PCOMPBIOL-D-23-00827R1 

A cell-and-plasma numerical model reveals hemodynamic stress and flow adaptation in zebrafish microvessels after morphological alteration

Dear Dr Phng,

I am pleased to inform you that your manuscript has been formally accepted for publication in PLOS Computational Biology. Your manuscript is now with our production department and you will be notified of the publication date in due course.

With kind regards,

Judit Kozma
